# Vision Transformer Adapter for Dense Predictions

**Zhe Chen**[1,2*], **Yuchen Duan**[2,3*], **Wenhai Wang**[2✉], **Junjun He**[2],
**Tong Lu**[1✉], **Jifeng Dai**[2,3], **Yu Qiao**[2]
[1]Nanjing University, [2]Shanghai AI Laboratory, [3]Tsinghua University
`czcz94cz@gmail.com`, `{duanyuchen,wangwenhai,hejunjun}@pjlab.org.cn`
`lutong@nju.edu.cn`, `{daijifeng,qiaoyu}@pjlab.org.cn`

## Abstract

This work investigates a simple yet powerful dense prediction task adapter for Vision Transformer (ViT). Unlike recently advanced variants that incorporate vision-specific inductive biases into their architectures, the plain ViT suffers inferior performance on dense predictions due to weak prior assumptions. To address this issue, we propose the ViT-Adapter, which allows plain ViT to achieve comparable performance to vision-specific transformers. Specifically, the backbone in our framework is a plain ViT that can learn powerful representations from large-scale multi-modal data. When transferring to downstream tasks, a **pre-training-free adapter** is used to introduce the image-related inductive biases into the model, making it suitable for these tasks. We verify ViT-Adapter on multiple dense prediction tasks, including object detection, instance segmentation, and semantic segmentation. Notably, without using extra detection data, our ViT-Adapter-L yields state-of-the-art **60.9** box AP and **53.0** mask AP on COCO test-dev. We hope that the ViT-Adapter could serve as an alternative for vision-specific transformers and facilitate future research. Code and models will be released at `https://github.com/czczup/ViT-Adapter`.

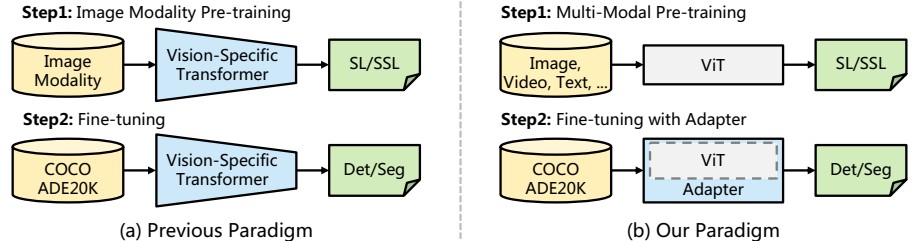

Figure 1: **Previous paradigm _vs._ our paradigm.** (a) Previous paradigm designs vision-specific models and pre-trains on large-scale image datasets via supervised or self-supervised learning and then fine-tunes them on downstream tasks. (b) We propose a pre-training-free adapter to close the performance gap between plain ViT (Dosovitskiy et al., 2020) and vision-specific transformers (_e.g._, Swin (Liu et al., 2021b)) for dense prediction tasks. Compared to the previous paradigm, our method preserves the flexibility of ViT and thus could benefit from advanced multi-modal pre-training.

## 1 Introduction

Recently, transformers have witnessed remarkable success in a broad range of computer vision fields. Benefiting from the dynamic modeling capability and the long-range dependence of the attention mechanism, various vision transformers (Dosovitskiy et al., 2020; Chen et al., 2021; Han et al., 2021; Li et al., 2021c; Wu et al., 2022b) soon rose in many computer vision tasks such as object detection and semantic segmentation, surpassing CNN models and reaching state-of-the-art performance. These models are mainly divided into two families, _i.e._ the plain ViT (Dosovitskiy et al., 2020; Touvron et al., 2021), and its hierarchical variants (Dong et al., 2021; Liu et al., 2021b; Wang et al., 2021; 2022a). In general, the latter can produce better results and is believed to introduce vision-specific inductive biases into their architectures by using local spatial operations.

---

[*]Equal contribution. [✉]Corresponding authors.

Nonetheless, the plain ViT (*i.e.*, vanilla transformer) still has some nonnegligible advantages. A typical example lies in multi-modal pre-training (Zhu et al., 2021; 2022; Wang et al., 2022b). Stemming from the natural language processing (NLP) field, transformer has no assumption of input data. Equipping with different tokenizers, *e.g.* patch embedding (Dosovitskiy et al., 2020), 3D patch embedding (Liu et al., 2021c), and token embedding (Vaswani et al., 2017), vanilla transformers such as *plain ViT can use massive multi-modal data for pre-training*, including image, video, and text, which encourages the model to learn semantic-rich representations. However, the plain ViT has conclusive defects in dense predictions compared to vision-specific transformers. Lacking image-related prior knowledge results in slower convergence and lower performance, and thus plain ViTs are hard to compete with vision-specific transformers (Huang et al., 2021b; Xie et al., 2021; Wang

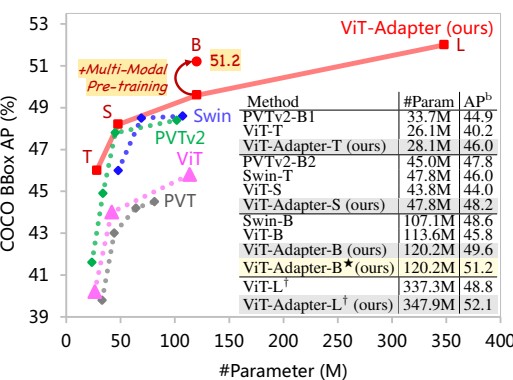

Figure 2: **Object detection performance on COCO val2017 using Mask R-CNN.** We see that the proposed ViT-Adapter brings significant improvements to plain ViTs. ★ indecates using multi-modal pre-trained ViT from (Zhu et al., 2021). Backbones pre-trained on ImageNet-22K are marked with [†], otherwise ImageNet-1K.

et al., 2022a) on dense prediction tasks. Inspired by the adapters (Houlsby et al., 2019; Stickland & Murray, 2019) in the NLP field, *this work aims to develop an adapter to close the performance gap between the plain ViT and vision-specific backbones for dense prediction tasks.*

To this end, we propose the Vision Transformer Adapter (ViT-Adapter), which is a *pre-training-free* additional network that can efficiently adapt the plain ViT to downstream dense prediction tasks without modifying its original architecture. Specifically, to introduce the vision-specific inductive biases into the plain ViT, we design three tailored modules for ViT-Adapter, including (1) a spatial prior module for capturing the local semantics (spatial prior) from input images, (2) a spatial feature injector for incorporating spatial prior into the ViT, and (3) a multi-scale feature extractor to reconstruct the multi-scale features required by dense prediction tasks.

As shown in Figure 1, compared to the previous paradigm that pre-trains on large-scale image datasets (*e.g.*, ImageNet (Deng et al., 2009)) then fine-tunes on other tasks, our paradigm is more flexible. In our framework, the backbone network is a general-propose model (*e.g.*, plain ViT) that can be pre-trained with not only images but also multi-modal data. For the transfer learning of dense prediction tasks, we use a randomly initialized adapter to introduce the image-related prior knowledge (inductive biases) into the pre-trained backbone, making the model suitable for these tasks. In this way, using ViT as the backbone, our framework achieves comparable or even better performance than vision-specific transformers such as Swin (Liu et al., 2021b).

Our main contributions are as follows:

- We explore a new paradigm to introduce vision-specific inductive biases into the plain ViT. It helps ViT achieve comparable performance to recent transformer variants (Liu et al., 2021b; Wang et al., 2022a) with regular ImageNet pre-training and further benefits from multi-modal pre-training.

- We design a spatial prior module and two feature interaction operations, to inject the image prior without redesigning the architecture of ViT. They can supplement the missing local information and reorganize fine-grained multi-scale features for dense prediction tasks.

- We evaluate the ViT-Adapter on multiple challenging benchmarks, including COCO (Lin et al., 2014) and ADE20K (Zhou et al., 2017). As shown in Figure 2, our models consistently achieve improved performance compared to the prior arts *under the fair pre-training strategy*. For instance, when using only ImageNet-1K pre-training, ViT-Adapter-B reports 49.6 box AP on COCO val, outperforming Swin-B by 1.0 points. Benefiting from multi-modal pre-training (Peng et al., 2022), our ViT-Adapter-L yields 60.9 box AP, which is the best record on COCO test-dev without training on extra detection data such as Objects365 (Shao et al., 2019).

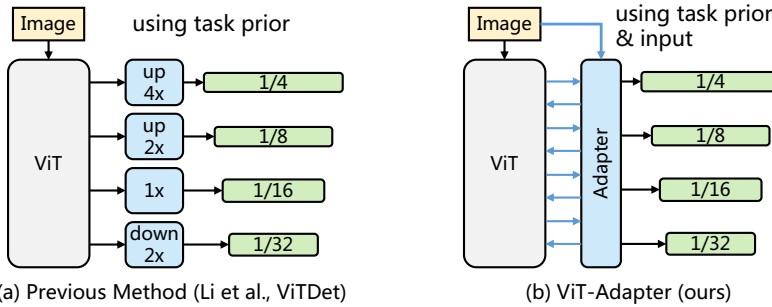

Figure 3: **Overview of ViT-Adapter and two related approaches.** Li et al. (2021b) and ViTDet (Li et al., 2022b) build simple feature pyramid to adapt plain ViT for object detection, which only consider task prior. Differently, our adapter utilizes both task prior and the input image.

## 2   RELATED WORK

**Transformers.** In recent years, transformers have dominated various tasks across multiple modalities, such as natural language processing, computer vision, and speech recognition. The vanilla transformer (Vaswani et al., 2017) was initially proposed for machine translation and remains the state-of-the-art architecture for NLP tasks today. ViT (Dosovitskiy et al., 2020) is the first work to generalize the vanilla transformer to the image classification task without much modification. PVT (Wang et al., 2021) and Swin (Liu et al., 2021b) introduce more vision-specific inductive biases by incorporating the pyramid structure from CNNs. Afterward, Conformer (Peng et al., 2021) proposed the first dual network to combine CNN with transformer. Recently, BEiT (Bao et al., 2022) and MAE (He et al., 2021) extended the scope of ViT to self-supervised learning with masked image modeling (MIM), demonstrating the powerful potential of the plain ViT architecture. Many works (Li et al., 2021b; Zhu et al., 2021; 2022; Wang et al., 2022b) have shown that designing vision-specific models is an important direction, but the general-propose architectures (*e.g.*, plain ViT) are more flexible and essential for masked data modeling and multi-modal pre-training. Therefore, we develop a pre-training-free adapter to introduce the image prior without modifying the architecture of ViT, preserving its flexibility and enjoying advanced multi-modal pre-training.

**Decoders for ViT.** The architecture for dense prediction commonly follows an encoder-decoder pattern, in which the encoder generates rich features and the decoder aggregates and translates them to the final predictions. Recently, illuminated by the global receptive fields of ViT, many works employ it as the encoder and design task-specific decoders. SETR (Zheng et al., 2021) is the first work to adopt ViT as the backbone and develop several CNN decoders for semantic segmentation. Segmenter (Strudel et al., 2021) also extends ViT to semantic segmentation, but differs in that it equips a transformer-based decoder. DPT (Ranftl et al., 2021) further applies ViT to the monocular depth estimation task via a CNN decoder and yields remarkable improvements. In summary, these works improve the dense prediction performance of ViT by designing modality- and task-specific decoders, but remain ViT's weakness of single-scale and low-resolution representation.

**Adapters.** To date, adapters have been widely used in the NLP field. PALs (Stickland & Murray, 2019) and Adapters (Houlsby et al., 2019) introduce new modules in transformer encoders for task-specific fine-tuning, making the pre-trained model quickly adapt to downstream NLP tasks. In the field of computer vision, some adapters have been proposed for incremental learning (Rosenfeld & Tsotsos, 2018) and domain adaptation (Rebuffi et al., 2017; 2018). With the advent of CLIP (Radford et al., 2021), many CLIP-based adapters (Gao et al., 2021; Sung et al., 2021; Zhang et al., 2021) were presented to transfer pre-trained knowledge to zero-shot or few-shot downstream tasks. Recently, Li et al. (2021b) and ViTDet (Li et al., 2022b) employed some upsampling and downsampling modules to adapt the plain ViT for object detection, as shown in Figure 3(a). However, under regular training settings (*i.e.*, apply ImageNet supervised pre-training and fine-tune for 36 epochs), their detection performance is still inferior[1] to recent models (Chu et al., 2021b; Dong et al., 2021; Wang et al., 2022a; Wu et al., 2022b) that well combine image prior. Therefore, it is still challenging to design a powerful dense prediction task adapter for ViT.

---

[1]In ViTDet, using regular ImageNet-22K pre-training instead of MAE (He et al., 2021) drops 4.0 box AP.

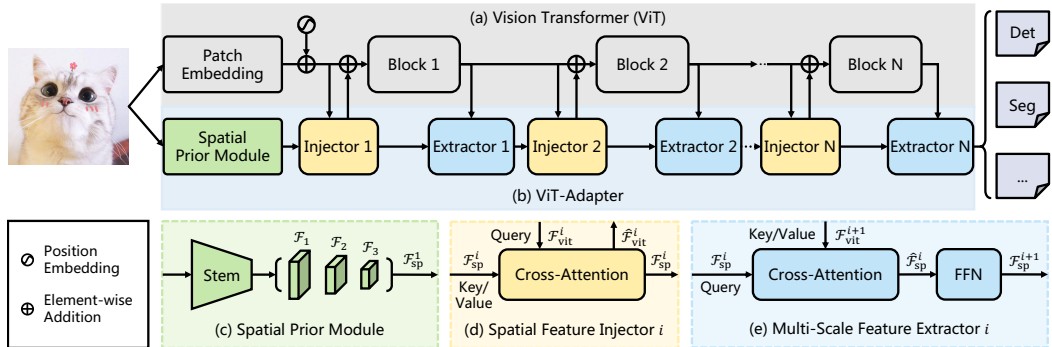

Figure 4: **Overall architecture of ViT-Adapter.** (a) The ViT, whose encoder layers are divided into $N$ (usually $N = 4$) equal blocks for feature interaction. (b) Our ViT-Adapter, which contains three key designs, including (c) a spatial prior module for modeling local spatial contexts from the input image, (d) a spatial feature injector for introducing spatial priors into ViT, and (e) a multi-scale feature extractor for reorganizing multi-scale features from the single-scale features of ViT.

## 3 VISION TRANSFORMER ADAPTER

### 3.1 OVERALL ARCHITECTURE

As illustrated in Figure 4, our model can be divided into two parts. The first part is the plain ViT (Dosovitskiy et al., 2020) that consists of a patch embedding followed by $L$ transformer encoder layers (see Figure 4(a)). The second part is the proposed ViT-Adapter as shown in Figure 4(b), which contains (1) a spatial prior module to capture spatial features from the input image, (2) a spatial feature injector to inject spatial priors into the ViT, and (3) a multi-scale feature extractor to extract hierarchical features from the single-scale features of ViT.

For the ViT, the input image is first fed into the patch embedding, where the image is divided into $16 \times 16$ non-overlapping patches. After that, these patches are flattened and projected to $D$-dimensional tokens, and the feature resolution is reduced to 1/16 of the original image. Then, these tokens added with the position embedding, are passed through $L$ encoder layers.

For the ViT-Adapter, we first feed the input image into the spatial prior module. $D$-dimensional spatial features of three target resolutions (*i.e.*, 1/8, 1/16, and 1/32) will be collected. Then, these feature maps are flattened and concatenated as the input for feature interaction. Specifically, given the number of interactions $N$ (usually $N = 4$), we evenly split the transformer encoders of ViT into $N$ blocks, each containing $L/N$ encoder layers. For the $i$-th block, we first inject spatial priors $\mathcal{F}_{\text{sp}}^i$ into the block via a spatial feature injector, and then extract hierarchical features from the output of the block by a multi-scale feature extractor. After $N$ feature interactions, we obtain high-quality multi-scale features, and then we split and reshape the features into three target resolutions 1/8, 1/16, and 1/32. Finally, we build the 1/4-scale feature map by upsampling the 1/8-scale feature map using a $2 \times 2$ transposed convolution. In this way, we obtain a feature pyramid of similar resolutions to ResNet (He et al., 2016), which can be used in various dense prediction tasks.

### 3.2 SPATIAL PRIOR MODULE

Recent studies (Wang et al., 2022a; Wu et al., 2021; Fang et al., 2022; Park & Kim, 2022) show convolutions can help transformers better capture the local spatial information. Inspired by this, we introduce the *Spatial Prior Module* (SPM). It is designed to model the local spatial contexts of images parallel with the patch embedding layer, so as not to alter the original architecture of ViT.

As shown in Figure 4(c), a standard convolutional stem borrowed from ResNet (He et al., 2016) is employed, which consists of three convolutions and a max-pooling layer. Then, we use a stack of stride-2 $3 \times 3$ convolutions to double the number of channels and reduce the size of feature maps. Finally, several $1 \times 1$ convolutions are applied at the end to project the feature maps to $D$ dimensions. In this way, we obtain a feature pyramid $\{\mathcal{F}_1, \mathcal{F}_2, \mathcal{F}_3\}$, which contains $D$-dimensional feature maps

with resolutions of 1/8, 1/16, and 1/32. Then, we flatten and concatenate these feature maps into feature tokens $\mathcal{F}_{sp}^1 \in \mathbb{R}^{(\frac{HW}{8^2} + \frac{HW}{16^2} + \frac{HW}{32^2}) \times D}$, as the input for feature interaction.

## 3.3 FEATURE INTERACTION

Due to weak prior assumptions, the plain ViT suffers sub-optimal performance on dense prediction tasks compared to vision-specific transformers (Chu et al., 2021a; Dong et al., 2021; Liu et al., 2021b; Wang et al., 2022a). To alleviate this issue, we propose two feature interaction modules to bridge the feature maps of our SPM and the ViT. To be specific, the two modules are mainly based on cross-attention, namely *Spatial Feature Injector* and *Multi-Scale Feature Extractor*.

**Spatial Feature Injector.** As shown in Figure 4(d), this module is used to inject the spatial priors into ViT. Specifically, for the $i$-th block of the ViT, we take the input feature $\mathcal{F}_{vit}^i \in \mathbb{R}^{\frac{HW}{16^2} \times D}$ as the query, and the spatial feature $\mathcal{F}_{sp}^i \in \mathbb{R}^{(\frac{HW}{8^2} + \frac{HW}{16^2} + \frac{HW}{32^2}) \times D}$ as the key and value. We use cross-attention to inject spatial feature $\mathcal{F}_{sp}^i$ into the input feature $\mathcal{F}_{vit}^i$, which can be written as Eqn. 1.

$$\hat{\mathcal{F}}_{vit}^i = \mathcal{F}_{vit}^i + \gamma^i \text{Attention}(\text{norm}(\mathcal{F}_{vit}^i), \text{norm}(\mathcal{F}_{sp}^i)), \tag{1}$$

where the $\text{norm}(\cdot)$ is LayerNorm (Ba et al., 2016), and the attention layer $\text{Attention}(\cdot)$ suggests using sparse attention. In addition, we apply a learnable vector $\gamma^i \in \mathbb{R}^D$ to balance the attention layer's output and the input feature $\mathcal{F}_{vit}^i$, which is initialized with $\mathbf{0}$. This initialization strategy ensures that the feature distribution of $\hat{\mathcal{F}}_{vit}^i$ will not be modified drastically due to the injection of spatial priors, thus making better use of the pre-trained weights of ViT.

**Multi-Scale Feature Extractor.** After injecting the spatial priors into the ViT, we obtain the output feature $\mathcal{F}_{vit}^{i+1}$ by passing $\hat{\mathcal{F}}_{vit}^i$ through the encoder layers of the $i$-th block. Then, we apply a module consisting of a cross-attention layer and a feed-forward network (FFN), to extract multi-scale features, as shown in Figure 4(e). This process can be formulated as:

$$\mathcal{F}_{sp}^{i+1} = \hat{\mathcal{F}}_{sp}^i + \text{FFN}(\text{norm}(\hat{\mathcal{F}}_{sp}^i)), \tag{2}$$

$$\hat{\mathcal{F}}_{sp}^i = \mathcal{F}_{sp}^i + \text{Attention}(\text{norm}(\mathcal{F}_{sp}^i), \text{norm}(\mathcal{F}_{vit}^{i+1})), \tag{3}$$

in which we use the spatial feature $\mathcal{F}_{sp}^i \in \mathbb{R}^{(\frac{HW}{8^2} + \frac{HW}{16^2} + \frac{HW}{32^2}) \times D}$ as the query, and the output feature $\mathcal{F}_{vit}^{i+1} \in \mathbb{R}^{\frac{HW}{16^2} \times D}$ as the key and value for cross-attention. As same as the spatial feature injector, we adopt sparse attention here to reduce computational cost. The generated spatial feature $\mathcal{F}_{sp}^{i+1}$ will be used as the input of the next spatial feature injector.

## 3.4 ARCHITECTURE CONFIGURATIONS

We build our ViT-Adapter for 4 different sizes of ViT, including ViT-T, ViT-S, ViT-B, and ViT-L. For these models, the parameter numbers of our adapters are 2.5M, 5.8M, 14.0M, and 23.7M, respectively. We employ deformable attention (Zhu et al., 2020) as the default sparse attention in our method, where the number of sampling points is fixed to 4, and the number of attention heads is set to 6, 6, 12, and 16. The number of interactions $N$ is 4, and in the last feature interaction, we stack three multi-scale feature extractors. Besides, we set the FFN ratio in our adapter to 0.25 to save computational overhead, *i.e.* the hidden sizes of FFN are 48, 96, 192, and 256 for 4 different adapters. More details of each configuration are shown in Table 10 in Appendix B.

## 4 EXPERIMENTS

Previous work (Wang et al., 2021) has shown that the pyramid prior is beneficial to dense prediction, but brings little gains to image classification. Therefore, in this study, we focus on how to better adapt readily available pre-trained ViTs to dense prediction tasks. We hope this method will also help decouple the model design of upstream pre-training and downstream fine-tuning.

### 4.1 OBJECT DETECTION AND INSTANCE SEGMENTATION

**Settings.** Our detection experiments are based on MMDetection (Chen et al., 2019b) and the COCO (Lin et al., 2014) dataset. We use 4 mainstream detectors to evaluate our ViT-Adapter, including

| Method | #Param (M) | Mask R-CNN 1× schedule | | | | | | Mask R-CNN 3×+MS schedule | | | | | |
|---|---|---|---|---|---|---|---|---|---|---|---|---|---|
| | | $AP^b$ | $AP^b_{50}$ | $AP^b_{75}$ | $AP^m$ | $AP^m_{50}$ | $AP^m_{75}$ | $AP^b$ | $AP^b_{50}$ | $AP^b_{75}$ | $AP^m$ | $AP^m_{50}$ | $AP^m_{75}$ |
| PVT-Tiny (Wang et al., 2021) | 32.9 | 36.7 | 59.2 | 39.3 | 35.1 | 56.7 | 37.3 | 39.8 | 62.2 | 43.0 | 37.4 | 59.3 | 39.9 |
| PVTv2-B1 (Wang et al., 2022a) | 33.7 | 41.8 | 64.3 | 45.9 | 38.8 | 61.2 | 41.6 | 44.9 | 67.3 | 49.4 | 40.8 | 64.0 | 43.8 |
| ViT-T (Li et al., 2021b) | 26.1 | 35.5 | 58.1 | 37.8 | 33.5 | 54.9 | 35.1 | 40.2 | 62.9 | 43.5 | 37.0 | 59.6 | 39.0 |
| ViTDet-T (Li et al., 2022b) | 26.6 | 35.7 | 57.7 | 38.4 | 33.5 | 54.7 | 35.2 | 40.4 | 63.3 | 43.9 | 37.1 | 60.1 | 39.3 |
| ViT-Adapter-T (ours) | 28.1 | 41.1 | 62.5 | 44.3 | 37.5 | 59.7 | 39.9 | 46.0 | 67.6 | 50.4 | 41.0 | 64.4 | 44.1 |
| PVT-Small (Wang et al., 2021) | 44.1 | 40.4 | 62.9 | 43.8 | 37.8 | 60.1 | 40.3 | 43.0 | 65.3 | 46.9 | 39.9 | 62.5 | 42.8 |
| PVTv2-B2 (Wang et al., 2022a) | 45.0 | 45.3 | 67.1 | 49.6 | 41.2 | 64.2 | 44.4 | 47.8 | 69.7 | 52.6 | 43.1 | 66.8 | 46.7 |
| Swin-T (Liu et al., 2021b) | 47.8 | 42.7 | 65.2 | 46.8 | 39.3 | 62.2 | 42.2 | 46.0 | 68.1 | 50.3 | 41.6 | 65.1 | 44.9 |
| ConvNeXt-T (Liu et al., 2022) | 48.1 | 44.2 | 66.6 | 48.3 | 40.1 | 63.3 | 42.8 | 46.2 | 67.9 | 50.8 | 41.7 | 65.0 | 44.9 |
| Focal-T (Yang et al., 2021) | 48.8 | 44.8 | 67.7 | 49.2 | 41.0 | 64.7 | 44.2 | 47.2 | 69.4 | 51.9 | 42.7 | 66.5 | 45.9 |
| ViT-S (Li et al., 2021b) | 43.8 | 40.2 | 63.1 | 43.4 | 37.1 | 59.9 | 39.3 | 44.0 | 66.9 | 47.8 | 39.9 | 63.4 | 42.2 |
| ViTDet-S (Li et al., 2022b) | 45.7 | 40.6 | 63.3 | 43.5 | 37.1 | 60.0 | 38.8 | 44.5 | 66.9 | 48.4 | 40.1 | 63.6 | 42.5 |
| ViT-Adapter-S (ours) | 47.8 | 44.7 | 65.8 | 48.3 | 39.9 | 62.5 | 42.8 | 48.2 | 69.7 | 52.5 | 42.8 | 66.4 | 45.9 |
| PVTv2-B5 (Wang et al., 2022a) | 101.6 | 47.4 | 68.6 | 51.9 | 42.5 | 65.7 | 46.0 | 48.4 | 69.2 | 52.9 | 42.9 | 66.6 | 46.2 |
| Swin-B (Liu et al., 2021b) | 107.1 | 46.9 | - | - | 42.3 | - | - | 48.6 | 70.0 | 53.4 | 43.3 | 67.1 | 46.7 |
| ViT-B (Li et al., 2021b) | 113.6 | 42.9 | 65.7 | 46.8 | 39.4 | 62.6 | 42.0 | 45.8 | 68.2 | 50.1 | 41.3 | 65.1 | 44.4 |
| ViTDet-B (Li et al., 2022b) | 121.3 | 43.2 | 65.8 | 46.9 | 39.2 | 62.7 | 41.4 | 46.3 | 68.6 | 50.5 | 41.6 | 65.3 | 44.5 |
| ViT-Adapter-B (ours) | 120.2 | 47.0 | 68.2 | 51.4 | 41.8 | 65.1 | 44.9 | 49.6 | 70.6 | 54.0 | 43.6 | 67.7 | 46.9 |
| ViT-L† (Li et al., 2021b) | 337.3 | 45.7 | 68.9 | 49.4 | 41.5 | 65.6 | 44.6 | 48.3 | 70.4 | 52.9 | 43.4 | 67.9 | 46.6 |
| ViTDet-L† (Li et al., 2022b) | 350.9 | 46.2 | 69.2 | 50.3 | 41.4 | 65.8 | 44.1 | 49.1 | 71.5 | 53.8 | 44.0 | 68.5 | 47.6 |
| ViT-Adapter-L† (ours) | 347.9 | 48.7 | 70.1 | 53.2 | 43.3 | 67.0 | 46.9 | 52.1 | 73.8 | 56.5 | 46.0 | 70.5 | 49.7 |

Table 1: **Object detection and instance segmentation with Mask R-CNN on COCO val2017.** For fair comparison, we initialize all ViT-T/S/B models with the regular ImageNet-1K pre-training (Touvron et al., 2021), and ViT-L† with the ImageNet-22K weights from (Steiner et al., 2021).

| Method | $AP^b$ | $AP^b_{50}$ | $AP^b_{75}$ | #P | Method | $AP^b$ | $AP^b_{50}$ | $AP^b_{75}$ | #P |
|---|---|---|---|---|---|---|---|---|---|
| Cascade Mask R-CNN 3×+MS schedule | | | | | ATSS 3×+MS schedule | | | | |
| Swin-T (Liu et al., 2021b) | 50.5 | 69.3 | 54.9 | 86M | Swin-T (Liu et al., 2021b) | 47.2 | 66.5 | 51.3 | 36M |
| Shuffle-T (Huang et al., 2021b) | 50.8 | 69.6 | 55.1 | 86M | Focal-T (Yang et al., 2021) | 49.5 | 68.8 | 53.9 | 37M |
| PVTv2-B2 (Wang et al., 2022a) | 51.1 | 69.8 | 55.3 | 83M | PVTv2-B2 (Wang et al., 2022a) | 49.9 | 69.1 | 54.1 | 33M |
| Focal-T (Yang et al., 2021) | 51.5 | 70.6 | 55.9 | 87M | ViT-S (Li et al., 2021b) | 45.2 | 64.8 | 49.0 | 32M |
| ViT-S (Li et al., 2021b) | 47.9 | 67.1 | 51.7 | 82M | ViT-Adapter-S (ours) | 49.6 | 68.5 | 54.0 | 36M |
| ViT-Adapter-S (ours) | 51.5 | 70.1 | 55.8 | 86M | GFL 3×+MS schedule | | | | |
| Swin-B (Liu et al., 2021b) | 51.9 | 70.9 | 57.0 | 145M | Swin-T (Liu et al., 2021b) | 47.6 | 66.8 | 51.7 | 36M |
| Shuffle-B (Huang et al., 2021b) | 52.2 | 71.3 | 57.0 | 145M | PVTv2-B2 (Wang et al., 2022a) | 50.2 | 69.4 | 54.7 | 33M |
| ViT-B (Li et al., 2021b) | 50.1 | 69.3 | 54.3 | 151M | ViT-S (Li et al., 2021b) | 46.0 | 65.5 | 49.7 | 32M |
| ViT-Adapter-B (ours) | 52.1 | 70.6 | 56.5 | 158M | ViT-Adapter-S (ours) | 50.0 | 69.1 | 54.3 | 36M |

Table 2: **Object detection with different frameworks on COCO val2017.** For fair comparison, we initialize all ViT-S/B models with the regular ImageNet-1K pre-training (Touvron et al., 2021). "#P" denotes the number of parameters. "MS" means multi-scale training.

Mask R-CNN (He et al., 2017), Cascade Mask R-CNN (Cai & Vasconcelos, 2019), ATSS (Zhang et al., 2020), and GFL (Li et al., 2020). To save time and memory, we refer to (Li et al., 2021b) and modify the $L$-layer ViT to use 14×14 window attention except for layers spaced at an interval of $L/4$. Following common practices (Wang et al., 2021), we adopt 1× or 3× training schedule (*i.e.*, 12 or 36 epochs) with a batch size of 16, and AdamW (Loshchilov & Hutter, 2017) optimizer with an initial learning rate of $1 \times 10^{-4}$ and a weight decay of 0.05.

**Results with ImageNet-1K Pre-training.** In Table 1 and Table 2, we apply the DeiT (Touvron et al., 2021) released ImageNet-1K weights (without distillation) as the initialization for all ViT-T/S/B models. We compare our ViT-Adapter with two related approaches (Li et al., 2021b; 2022b) and multiple representative vision-specific backbones (Wang et al., 2021; 2022a; Huang et al., 2021b; Liu et al., 2021b; Yang et al., 2021). As we can see, when using regular training settings for fair comparison, the detection performance of ViT (Li et al., 2021b) and ViTDet (Li et al., 2022b) is inferior to recent vision-specific models. For example, with Mask R-CNN and 3×+MS schedule,

| Method | Pre-train | Crop Size | Semantic FPN 80k | | | UperNet 160k | | |
|---|---|---|---|---|---|---|---|---|
| | | | #Param | mIoU | +MS | #Param | mIoU | +MS |
| PVT-Tiny (Wang et al., 2021) | IN-1K | 512×512 | 17.0M | 36.6 | 37.3 | 43.2M | 38.5 | 39.0 |
| ViT-T (Li et al., 2021b) | IN-1K | 512×512 | 10.2M | 39.4 | 40.5 | 34.1M | 41.7 | 42.6 |
| ViT-Adapter-T (ours) | IN-1K | 512×512 | 12.2M | 41.7 | 42.1 | 36.1M | 42.6 | 43.6 |
| PVT-Small (Wang et al., 2021) | IN-1K | 512×512 | 28.2M | 41.9 | 42.3 | 54.5M | 43.7 | 44.0 |
| PVTv2-B2 (Wang et al., 2022a) | IN-1K | 512×512 | 29.1M | 45.2 | 45.7 | - | - | - |
| Swin-T (Liu et al., 2021b) | IN-1K | 512×512 | 31.9M | 41.5 | - | 59.9M | 44.5 | 45.8 |
| Twins-SVT-S (Chu et al., 2021a) | IN-1K | 512×512 | 28.3M | 43.2 | - | 54.4M | 46.2 | 47.1 |
| ViT-S (Li et al., 2021b) | IN-1K | 512×512 | 27.8M | 44.6 | 45.8 | 53.6M | 44.6 | 45.7 |
| ViT-Adapter-S (ours) | IN-1K | 512×512 | 31.9M | 46.1 | 46.6 | 57.6M | 46.2 | 47.1 |
| Swin-B (Liu et al., 2021b) | IN-1K | 512×512 | 91.2M | 46.0 | - | 121.0M | 48.1 | 49.7 |
| Twins-SVT-L (Chu et al., 2021a) | IN-1K | 512×512 | 103.7M | 46.7 | - | 133.0M | 48.8 | 50.2 |
| ViT-B (Li et al., 2021b) | IN-1K | 512×512 | 98.0M | 46.4 | 47.6 | 127.3M | 46.1 | 47.1 |
| ViT-Adapter-B (ours) | IN-1K | 512×512 | 104.6M | 47.9 | 48.9 | 133.9M | 48.8 | 49.7 |
| Swin-B[†] (Liu et al., 2021b) | IN-22K | 640×640 | - | - | - | 121.0M | 50.0 | 51.7 |
| Swin-L[†] (Liu et al., 2021b) | IN-22K | 640×640 | - | - | - | 234.0M | 52.1 | 53.5 |
| ViT-Adapter-B[†] (ours) | IN-22K | 512×512 | 104.6M | 50.7 | 51.9 | 133.9M | 51.9 | 52.5 |
| ViT-Adapter-L[†] (ours) | IN-22K | 512×512 | 332.0M | 52.9 | 53.7 | 363.8M | 53.4 | 54.4 |
| ViT-Adapter-L[★] (ours) | MM | 512×512 | 332.0M | 54.2 | 54.7 | 363.8M | 55.0 | 55.4 |

Table 3: **Semantic segmentation on the ADE20K val.** Semantic FPN (Kirillov et al., 2019) and UperNet (Xiao et al., 2018) are used as segmentation frameworks. "IN-1K/22K" and "MM" represent ImageNet-1K/22K and multi-modal pre-training, respectively. "MS" means multi-scale testing.

ViT-S and ViTDet-S are 3.8 $AP^b$ and 3.3 $AP^b$ lower than PVTv2-B2 (Wang et al., 2022a) respectively. Differently, our ViT-Adapter-S outperforms these two approaches by clear margins and even 0.4 $AP^b$ higher than PVTv2-B2. This observation can also be seen in the experiments of three other detectors, including Cascade Mask R-CNN, ATSS, and GFL. These results indicate that, *with only the regular ImageNet-1K pre-training*, ViT-Adapter can promote the plain ViT to attain similar or even superior performance than these vision-specific transformers.

**Results with ImageNet-22K Pre-training.** In Table 1, we employ the ImageNet-22K pre-trained weights from AugReg (Steiner et al., 2021) to initialize all ViT-L models, including ViT (Li et al., 2021b), ViTDet (Li et al., 2022b), and our ViT-Adapter. It can be seen that, when training Mask R-CNN with 3×+MS schedule, our ViT-Adapter-L[†] brings 3.8 $AP^b$ and 3.0 $AP^b$ improvements over ViT-L[†] (Li et al., 2021b) and ViTDet-L[†] (Li et al., 2022b), respectively.

**Results with Multi-Modal Pre-training.** In this experiment, we study the effect of multi-modal pre-training. Specifically, we fine-tune the ViT-Adapter-B with Mask R-CNN for the 3×+MS schedule using different pre-trained weights. As shown in Table 4, simply replacing the ImageNet-22K pre-training (Steiner et al., 2021) with the multi-modal pre-training (Zhu et al., 2021) gives us a significant gain of 0.7 $AP^b$ and $AP^m$. These results indicate that our method can easily derive considerable benefits from advanced multi-modal pre-training, which is difficult for vision-specific models like Swin.

| Method | Pre-train | $AP^b$ | $AP^m$ |
|---|---|---|---|
| Swin-B (Mask R-CNN 3×+MS) | ImageNet-1K | 48.6 | 43.3 |
| | ImageNet-22K | 49.6 | 44.3 |
| | Multi-Modal | N/A | N/A |
| ViT-Adapter-B (Mask R-CNN 3×+MS) | ImageNet-1K | 49.6 | 43.6 |
| | ImageNet-22K | 50.5 | 44.6 |
| | Multi-Modal | **51.2** | **45.3** |

Table 4: **Comparison of different pre-trained weights.** Our method retains the flexibility of ViT and thus could benefit from advanced multi-modal pre-training (Zhu et al., 2021).

## 4.2 SEMANTIC SEGMENTATION

**Settings.** We evaluate our ViT-Adapter on semantic segmentation with the ADE20K (Zhou et al., 2017) dataset and MMSegmentation (Contributors, 2020) codebase. Both Semantic FPN (Kirillov et al., 2019) and UperNet (Xiao et al., 2018) are employed as the basic frameworks. For Semantic FPN, we apply the settings of PVT (Wang et al., 2021) and train the models for 80k iterations. For UperNet, we follow the settings of Swin (Liu et al., 2021b) to train it for 160k iterations.

**Results with ImageNet-1K Pre-training.** In Table 3, we report the semantic segmentation results in terms of single-scale and multi-scale (MS) mIoU. As same as Section 4.1, we initialize all ViT-T/S/B models with the DeiT (Touvron et al., 2021) released ImageNet-1K weights. It shows that, under comparable model sizes, our method surpasses the ViT (Li et al., 2021b) and many representative vision-specific transformers (Wang et al., 2021; 2022a; Liu et al., 2021b; Chu et al., 2021a). For instance, our ViT-Adapter-S achieves 47.1 MS mIoU with UperNet, outperforming many strong counterparts such as Swin-T. Similarly, ViT-Adapter-B reports a competitive performance of 49.7 MS mIoU, which is 2.6 points higher than ViT-B and on par with Swin-B and Twins-SVT-L. These fair comparisons using only regular ImageNet-1K pre-training (Touvron et al., 2021) demonstrate the effectiveness and universality of our ViT-Adapter.

**Results with ImageNet-22K Pre-training.** When using the ImageNet-22K pre-trained weights (Steiner et al., 2021), our ViT-Adapter-B$^{\dagger}$ attains 51.9 mIoU and 52.5 MS mIoU with UperNet, exceeding Swin-B$^{\dagger}$ by at least 0.8 mIoU. Similarly, ViT-Adapter-L$^{\dagger}$ yields the results of 53.4 mIoU and 54.4 MS mIoU, which is outstanding from the counterparts like Swin-L$^{\dagger}$. These significant and consistent improvements over different model sizes suggest that our method can cover the shortage of plain ViT, making it more suitable for semantic segmentation.

**Results with Multi-Modal Pre-training.** Here, we apply the multi-modal pre-trained weights from Uni-Perceiver (Zhu et al., 2021) for semantic segmentation. As shown in Table 3, for Semantic FPN and UperNet, replacing the ImageNet-22K pre-training with multi-modal pre-training benefits our ViT-Adapter-L$^{\star}$ with impressive gains of 1.3 mIoU and 1.6 mIoU, respectively.

### 4.3 COMPARISONS WITH STATE-OF-THE-ARTS

**Settings.** We conduct experiments to combine our ViT-Adapter with state-of-the-art detection/segmentation frameworks, including HTC++ (Liu et al., 2021b) (without extra detection dataset) and Mask2Former (Cheng et al., 2021), and recent multi-modal pre-training BEiTv2 (Peng et al., 2022). The experimental settings are listed in Appendix A.1 and A.2.

**Results.** As shown in Table 5, our method reaches state-of-the-art performance. While these results may be partly due to the effectiveness of advanced pre-training, our study demonstrates that plain backbone detectors/segmenters can challenge the entrenched position of hierarchical backbones.

| COCO test-dev | $AP^b$ | $AP^m$ | ADE20K val | mIoU |
|---|---|---|---|---|
| CB-Swin-L | 60.1 | 52.3 | FD-SwinV2-G | 61.4 |
| SwinV2-L | 60.8 | 52.7 | **ViT-Adapter-L** | **61.5** |
| **ViT-Adapter-L** | **60.9** | **53.0** | BEiT3(w/ ViT-Adapter) | **62.8** |

Table 5: **Comparison with previous SOTA.**

### 4.4 ABLATION STUDY

**ViT *vs.* ViT-Adapter Feature.** Recent works (Park & Kim, 2022; Si et al., 2022) show that ViT presents the characteristics of learning low-frequency global signals, while CNN tends to extract high-frequency information (*e.g.*, local edges and textures). To show the difference between the features of ViT and ViT-Adapter, we first use Fourier analysis as a toolkit for visualization. As shown in Figure 5(a), the Fourier spectrum and relative log amplitudes of the Fourier transformed feature maps (average over 100 images) indicate that ViT-Adapter captures more high-frequency signals than the ViT (Li et al., 2021b) baseline. In addition, we also visualize the stride-8 feature map in Figure 5 (b)(c), which shows that the features of ViT are blurry and coarse. In contrast, our features are more fine-grained and have more local edges and textures. This observation demonstrates that our method grafts the merit of CNN for capturing high-frequency information to ViT.

**Ablation for Components.** To investigate the contribution of each key design, we gradually extend the ViT-S baseline (Li et al., 2021b) to our ViT-Adapter-S. All models are trained with Mask R-CNN for 1× schedule. As shown in the left side of Table 6, by directly resizing and adding the spatial features from SPM, our variant 1 improves 1.4 $AP^b$ and 0.9 $AP^m$ over the baseline, showing that local spatial information is essential for dense prediction. From variant 2, we find that the spatial feature injector further boosts the performance by 1.0 $AP^b$ and 0.8 $AP^m$. This observation illustrates that cross-attention is a more flexible way to inject spatial features. Moreover, we employ the multi-scale feature extractor to reconstruct hierarchical features, which brings 2.1 $AP^b$ and 1.1 $AP^m$ gains, alleviating ViT's drawback of single-scale features. In summary, our proposed components are each necessary and collectively create 4.5 $AP^b$ and 2.8 $AP^m$ improvements.

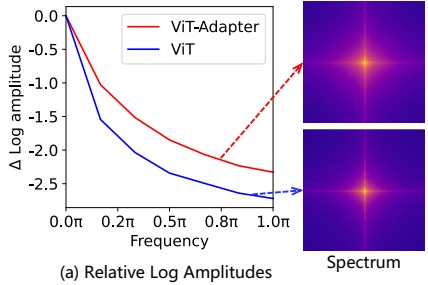
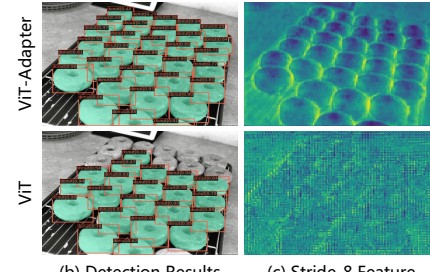

Figure 5: **ViT *vs.* ViT-Adapter Feature.** (a) Relative log amplitudes of Fourier transformed feature maps. (b) Detection results. (c) Stride-8 feature map. Compared to the ViT baseline (Li et al., 2021b), our ViT-Adapter captures more high-frequency signals, and produces more fine-grained features with rich edges and textures, which is of great help for dense prediction.

| Method | Components | | | Interaction Mode | Mask R-CNN 1× | | | | $N$ | $AP^b$ | $AP^m$ | #Param |
|---|---|---|---|---|---|---|---|---|---|---|---|---|
| | SPM | Injector | Extractor | | $AP^b$ | $AP^m$ | #Param | | 0 | 40.2 | 37.1 | 43.8M |
| ViT-S (Li et al., 2021b) | | | | - | 40.2 | 37.1 | 43.8M | | 1 | 43.2 | 38.9 | 45.5M |
| Variant 1 | ✓ | | | Add | 41.6 | 38.0 | 45.1M | | 2 | 43.9 | 39.4 | 46.2M |
| Variant 2 | ✓ | ✓ | | Attention | 42.6 | 38.8 | 46.6M | | 4 | **44.7** | **39.9** | 47.8M |
| ViT-Adapter-S (ours) | ✓ | ✓ | ✓ | Attention | **44.7** | **39.9** | 47.8M | | 6 | 44.7 | 39.8 | 49.4M |

Table 6: **Ablation studies of ViT-Adapter.** (Left) ablation of key components. Our proposed components collectively bring 4.5 $AP^b$ and 2.8 $AP^m$ gains. (Right) ablation of the number of interactions $N$. The model gives the best performance when $N = 4$. SPM is short for the spatial prior module.

| Attention Mechanism | Complexity | $AP^b$ | $AP^m$ | FLOPs | #Param | Train Time | Memory |
|---|---|---|---|---|---|---|---|
| Global Attention (Vaswani et al., 2017) | Quadratic | 43.7 | 39.3 | 1080G | 50.3M | 1.61s | *19.0G |
| CSwin Attention (Dong et al., 2021) | Linear | 43.5 | 39.2 | 456G | 50.3M | 0.56s | 15.6G |
| Pale Attention (Wu et al., 2022a) | Linear | 44.2 | 39.8 | 458G | 50.3M | 0.75s | 17.4G |
| Deformable Attention (Zhu et al., 2020) | Linear | **44.7** | **39.9** | **403G** | **47.8M** | **0.36s** | **13.7G** |

Table 7: **Ablation of using different attention mechanisms in our adapter.** The per-iteration training time and GPU training memory are measured by A100 GPUs with per-GPU batch size 2 and FP16 training. "*" indicates using activation checkpointing to save training memory.

**Number of Interactions.** In the right side of Table 6, we study the effect of the number of interactions. Specifically, we build several ViT-Adapter-S variants with different numbers of interactions. We observe that the model accuracy saturates when $N$ goes larger, and applying more interactions cannot monotonically promote the performance. Therefore, we empirically set $N$ to 4 by default.

**Attention Type.** Our method is a general framework in which the attention mechanism is replaceable. To verify this, we adopt ViT-Adapter-S as the basic model and study 4 different attention mechanisms. As shown in Table 7, sparse attention with linear complexity is more suitable for our adapter than global attention with quadratic complexity. We ended up using deformable attention (Zhu et al., 2020) as the default configuration. Notably, it can be replaced by other more advanced attention mechanisms in the future to further boost performance.

## 5 CONCLUSION

This work explores a new paradigm, namely ViT-Adapter, to bridge the performance gap between the plain ViT and vision-specific transformers on dense prediction tasks. Without modifying the inherent architecture, we flexibly inject image-related inductive biases into the ViT and reconstruct fine-grained multi-scale features required by dense predictions. Extensive experiments on object detection, instance segmentation, and semantic segmentation show that our method can achieve comparable or even better performance than well-designed vision-specific transformers, and further derive considerable benefits from advanced multi-modal pre-training.

ACKNOWLEDGEMENT

This work is partly supported by the National Natural Science Foundation of China (Grant No. 61672273, 61832008), and the Shanghai Committee of Science and Technology (Grant No. 21DZ1100100).

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

| Method | Framework | Epoch | Backbone Pre-train | val AP$^b$ | val AP$^m$ | val (+MS) AP$^b$ | val (+MS) AP$^m$ | test-dev AP$^b$ | test-dev AP$^m$ | test-dev (+MS) AP$^b$ | test-dev (+MS) AP$^m$ |
|---|---|---|---|---|---|---|---|---|---|---|---|
| Swin-L | HTC++ | 72 | IN-22K, sup | 57.1 | 49.5 | 58.0 | 50.4 | 57.7 | 50.2 | 58.7 | 51.1 |
| Focal-L | HTC++ | 36 | IN-22K, sup | 57.0 | 49.9 | 58.1 | 50.9 | - | - | 58.4 | 51.3 |
| MViTv2-L | Cascade | 50 | IN-22K, sup | 56.9 | 48.6 | 58.7 | 50.5 | - | - | - | - |
| MViTv2-H | Cascade | 50 | IN-22K, sup | 57.1 | 48.8 | 58.4 | 50.1 | - | - | - | - |
| CBV2-Swin-L | HTC | 36 | IN-22K, sup | 59.1 | 51.0 | 59.6 | 51.8 | 59.4 | 51.6 | 60.1 | 52.3 |
| ViT-Adapter-L | HTC++ | 36 | IN-22K, sup | 56.6 | 49.0 | 57.7 | 49.9 | 57.4 | 50.0 | 58.4 | 50.7 |
| Swin-L | HTC++ | 36 | IN-1K, UM-MAE | 57.4 | 49.8 | 58.7 | 50.9 | - | - | - | - |
| ViTDet-L | Cascade | 100 | IN-1K, MAE | 59.6 | 51.1 | 60.4 | 52.2 | - | - | - | - |
| ViT-Adapter-L | HTC++ | 36 | IN-22K, BEiT | 58.4 | 50.8 | 60.2 | 52.2 | 58.9 | 51.3 | 60.4 | 52.5 |
| ViT-Adapter-L | HTC++ | 36 | MM$^\dagger$, BEiTv2 | 58.8 | **51.1** | **60.5** | **52.5** | **59.5** | **51.8** | **60.9** | **53.0** |

Table 8: **Comparisons with the leading results on the COCO val2017 and test-dev sets.** There are three detection frameworks used, including Cascade Mask R-CNN (Cai & Vasconcelos, 2019), HTC (Chen et al., 2019a), and its extension HTC++ (Liu et al., 2021b). All of these models are trained *without* extra detection datasets, such as Objects365 (Shao et al., 2019). "IN-22K, sup" is short for ImageNet-22K supervised pre-training. "MS" indicates multi-scale testing. "$^\dagger$": Since the pre-trained CLIP (Radford et al., 2021) model is used in the training process of BEiTv2 (Peng et al., 2022), we regard it as a multi-modal pre-training method.

# A    COMPARISON WITH PREVIOUS STATE-OF-THE-ARTS

In recent years, the state-of-the-art models on dense prediction benchmarks are primarily vision-specific transformers, such as Swin (Liu et al., 2021b), Focal (Yang et al., 2021), MViTv2 (Li et al., 2021a), and SwinV2 (Liu et al., 2021a), while the plain ViT is rarely found. Nevertheless, we argue that the plain ViT still has the potential to reach the leading performance by leveraging our ViT-Adapter. To verify this, we conduct extensive additional experiments as follows.

## A.1    OBJECT DETECTION AND INSTANCE SEGMENTATION

**Settings.** Following prior art (Li et al., 2021b), we modify the 24-layer ViT-L to use $14 \times 14$ window attention except for layers spaced at an interval of 6, to save training time and memory. The state-of-the-art detector HTC++ (Liu et al., 2021b) is employed for our experiments. Specifically, we rescale the shorter side of images between 400 and 1400, while the longer side is at most 1600. Instaboost (Fang et al., 2019), Soft-NMS (Bodla et al., 2017), AdamW (Loshchilov & Hutter, 2017) optimizer (batch size of 16, initial learning rate of $1 \times 10^{-4}$, and weight decay of 0.05), and $3 \times$ schedule are adopted during training. We use a layer-wise learning rate decay of 0.9, and a drop path rate of 0.4. For a fairer comparison, here we take two initialization strategies, *i.e.* regular ImageNet-22K pre-training, and more advanced self-supervised or multi-modal pre-training.

**Results with ImageNet-22K Pre-training.** As shown in Table 8, with the ImageNet-22K supervised pre-training from AugReg (Steiner et al., 2021), our ViT-Adapter-L reports 58.4 AP$^b$ and 50.7 AP$^m$ on the COCO test-dev, which is comparable to many vision-specific transformers such as Swin-L (58.4 AP$^b$ *vs.* 58.7 AP$^b$) and Focal-L (58.4 AP$^b$ *vs.* 58.4 AP$^b$). This fair comparison illustrates that, our ViT-Adapter significantly narrows the performance gap between the plain ViT and well-designed vision-specific models.

**Results with More Advanced Pre-training.** Since our paradigm retains the flexibility of the plain ViT, it can *easily derive significant benefits from advanced pre-training techniques*, such as multi-modal pre-training (Zhu et al., 2021; 2022) or self-supervised pre-training (Bao et al., 2022; Peng et al., 2022; He et al., 2021), or a combination of the both (Wang et al., 2022b). Here, we take the readily available weights from BEiT (Bao et al., 2022) and BEiTv2 (Bao et al., 2022) as examples. Due to BEiT using learnable relative position biases instead of the absolute position embeddings, we replace the remaining global attention (see the settings part) with $56 \times 56$ window attention as an approximation. For these layers, the relative position biases need to be interpolated to adapt to the new window size.

| Method | Framework | Backbone Pre-train | Extra Pre-train | Crop Size | Iters | ADE20K val mIoU | +MS | #Param |
|--------|-----------|--------------------|-----------------|-----------|-------|-----------------|-----|--------|
| Swin-L | Mask2Former | IN-22K, sup | - | 640 | 160k | 56.1 | 57.3 | 215M |
| Swin-L-FaPN | Mask2Former | IN-22K, sup | - | 640 | 160k | 56.4 | 57.7 | 217M |
| SeMask-Swin-L | Mask2Former | IN-22K, sup | - | 640 | 160k | 57.0 | 58.2 | - |
| HorNet-L | Mask2Former | IN-22K, sup | - | 640 | 160k | 57.5 | 57.9 | - |
| ViT-Adapter-L | Mask2Former | IN-22K, sup | - | 640 | 160k | 56.8 | 57.7 | 438M |
| BEiT-L | UperNet | IN-22K, BEiT | - | 640 | 160k | 56.7 | 57.0 | 441M |
| ViT-Adapter-L | UperNet | IN-22K, BEiT | - | 640 | 160k | 58.0 | 58.4 | 451M |
| BEiTv2-L | UperNet | IN-22K, BEiTv2 | - | 512 | 160k | 57.5 | 58.0 | 441M |
| ViT-Adapter-L | UperNet | IN-22K, BEiTv2 | - | 512 | 160k | 58.0 | 58.5 | 451M |
| ConvNeXt-XL* | Mask2Former | IN-22K, sup | COCO-Stuff, sup | 896 | 80k | 57.1 | 58.4 | 588M |
| Swin-L* | Mask2Former | IN-22K, sup | COCO-Stuff, sup | 896 | 80k | 57.3 | 58.3 | 434M |
| SwinV2-G | UperNet | IN-22K, sup | Ext-70M, sup | 896 | 160k | 59.3 | 59.9 | 3.0B |
| FD-SwinV2-G | UperNet | IN-22K, sup | Ext-70M, sup | 896 | 160k | - | 61.4 | 3.0B |
| Swin-L | Mask DINO | IN-22K, sup | Objects365, sup | - | 160k | 59.5 | 60.8 | 223M |
| ViT-Adapter-L | Mask2Former | IN-22K, BEiT | COCO-Stuff, sup | 896 | 80k | 59.4 | 60.5 | 571M |
| ViT-Adapter-L | Mask2Former | MM[†], BEiTv2 | COCO-Stuff, sup | 896 | 80k | **61.2** | **61.5** | 571M |
| BEiT-3 (w/ ViT-Adapter) | Mask2Former | MM, BEiT-3 | COCO-Stuff, sup | 896 | 80k | **62.0** | **62.8** | 1.3B |

Table 9: **Comparison with previous state-of-the-art results on the ADE20K validation set.** The results of BEiT-3 are collected from (Wang et al., 2022b). "*": We follow BEiT (Bao et al., 2022) to use a wider segmentation head for ConvNeXt-XL and Swin-L to match the number of parameters, and apply the same training strategy to them. "IN-22K, sup": ImageNet-22K supervised pre-training. "MM": Multi-modal pre-training. "MS": Multi-scale testing. "[†]": Since the pre-trained CLIP (Radford et al., 2021) model is used in the training process of BEiTv2 (Peng et al., 2022), we regard it as a multi-modal pre-training method.

As reported in Table 8, our ViT-Adapter-L (w/ BEiT) creates 60.4 AP$^b$ and 52.5 AP$^m$ on the COCO test-dev, and ViT-Adapter-L (w/ BEiTv2) further sets this record to 60.9 AP$^b$ and 53.0 AP$^m$. Notably, although it's not a perfectly controlled comparison, our method attains similar performance taking fewer training epochs (36 *vs.* 100) than ViTDet (Li et al., 2022b). We argue that a longer training schedule such as 100 epochs may bring an added bonus, but it is expensive to afford due to limited computing resources. In summary, from a system-level perspective, our ViT-Adapter can enjoy the dividends of various advanced pre-training techniques and help plain ViT achieve leading performance on the object detection and instance segmentation tasks.

## A.2 SEMANTIC SEGMENTATION

**Settings.** For semantic segmentation, we employ the AdamW optimizer with an initial learning rate of $2 \times 10^{-5}$, a batch size of 16, and a weight decay of 0.05. Layer-wise learning rate decay of 0.9 and drop path rate of 0.4 are used to train the models. Other training settings, such as pre-training techniques, crop size, and the number of iterations, are listed in Table 9.

**Results with ImageNet-22K Pre-training.** As shown in Table 9, using Mask2Former (Cheng et al., 2021) as the segmenter, our ViT-Adapter-L achieves 56.8 mIoU and 57.7 MS mIoU on the ADE20K val, which is comparable to recent vision-specific models, such as Swin-L (Liu et al., 2021b), Swin-L-FaPN (Huang et al., 2021a), SeMask-Swin-L (Jain et al., 2021), and HorNet-L (Rao et al., 2022).

**Results with More Advanced Pre-training.** It can be seen from Table 9, when training with UperNet for 160k iterations, our ViT-Adapter-L (w/ BEiT) yields 58.4 MS mIoU, outperforming BEiT-L by 1.4 points with only 10M additional parameters. It shows that our adapter can deliver significant benefits even for a powerful self-supervised pre-trained ViT.

Furthermore, we compare the performance of our method with vision-specific models that also use additional datasets. For example, SwinV2-G (Liu et al., 2021a) uses a privately collected ImageNet-22K-ext-70M dataset that contains 70 million images. Mask DINO (Li et al., 2022a) takes the detection pre-training on large-scale Objects365 (Shao et al., 2019) dataset as the initialization for segmentation. Due to limited computing resources, we explore a *simple and affordable* transfer

| Variants | Settings of ViT | | | | | Settings of Adapter | | | | Total |
| | Layers | Width | FFN | Heads | #Param | $N$ | FFN | Heads | #Param | Param |
| --- | --- | --- | --- | --- | --- | --- | --- | --- | --- | --- |
| Tiny (T) | 12 | 192 | 768 | 3 | 5.5M | 4 | 48 | 6 | 2.5M | 8.0M |
| Small (S) | 12 | 384 | 1536 | 6 | 21.7M | 4 | 96 | 6 | 5.8M | 27.5M |
| Base (B) | 12 | 768 | 3072 | 12 | 85.8M | 4 | 192 | 12 | 14.0M | 99.8M |
| Large (L) | 24 | 1024 | 4096 | 16 | 303.3M | 4 | 256 | 16 | 23.7M | 327.0M |

Table 10: **Configurations of the ViT-Adapter.** We apply our adapters on four different settings of ViT, including ViT-T, ViT-S, ViT-B, and ViT-L, covering a wide range of different model sizes.

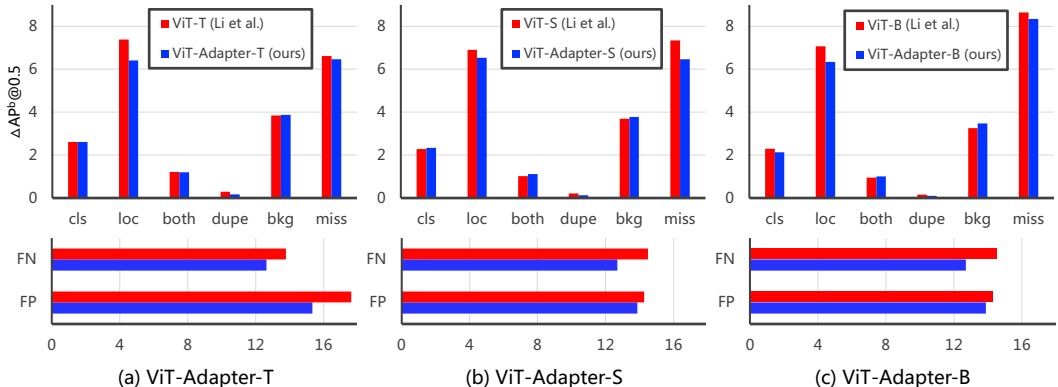

Figure 6: **TIDE error type analysis (the lower the better).** We use the models listed in Table 1 for analysis. As defined in (Bolya et al., 2020), we plot the AP$^b$ metric at an IoU threshold of 0.5. These bars show the effect of each error type on overall detection performance). The error types include: **cls**: localized correctly but classified incorrectly; **loc**: classified correctly but localized incorrectly; **both**: classified incorrectly and localized incorrectly; **dupe**: detection would be correct if not for a higher scoring detection; **bkg**: detected background as foreground; **miss**: all undetected ground-truth not covered by other error types; **FN**: false negatives; **FP**: false positives. We observe that our ViT-Adapter makes fewer localization and miss errors than the ViT baseline (Li et al., 2021b), and occurs fewer false positive and negative errors.

learning strategy for semantic segmentation. Specifically, we use the COCO-Stuff (Caesar et al., 2018) dataset for 80k iterations of pre-training, and then ADE20K for 80k iterations of fine-tuning. The total number of iterations is still 160k, and no additional training overhead is added.

Under this setting, our ViT-Adapter-L (w/ BEiT) produces an exciting score of 60.5 MS mIoU. Further, ViT-Adapter-L (w/ BEiTv2) creates a new record of 61.5 MS mIoU, which is slightly better than FD-SwinV2-G (Wei et al., 2022), while the parameter number is much smaller (571M *vs.* 3.0B). It's worth noting that, our ViT-Adapter is also adopted by the recently proposed BEiT-3 (Wang et al., 2022b), which is a ViT-style foundation model that can be pre-trained with multi-modal data. As described in their paper, using ViT-Adapter for the transfer learning of semantic segmentation, BEiT-3 establishes a new state-of-the-art of 62.8 MS mIoU on ADE20K val, which is a convincing verification of the paradigm we present in Figure 1.

# B   ADDITIONAL ABLATION AND DISCUSSION

**Architecture Configurations.** The more detailed configurations are listed in Table 10.

**TIDE Error Type Analysis.** TIDE (Bolya et al., 2020) is a toolbox for analyzing the sources of error in object detection algorithms. Following (Li et al., 2021b), we show the error type analysis in Figure 6. For fair comparison, the models listed in Table 1 are adopted for analysis. These results reveal where our ViT-Adapter improves overall AP$^b$ relative to the ViT baseline (Li et al., 2021b). For instance, we observe that our adapter helps reduce missed and localization errors, and has a substantial effect on fixing false negative and positive errors.

**Feature Visualization.** We plot more visualization of feature maps produced by ViT-B (Li et al., 2021b) and our ViT-Adapter-B in Figure 7 and Figure 8, which are trained based on Mask R-CNN for detection and UperNet for segmentation, respectively. As can be seen, the features of ViT-B are blurry and coarse, while our features are more refined and have more local edges and textures. This observation also accords with the Fourier analysis in Section 4.4, which demonstrates that ViT has the characteristics of capturing low-frequency information, and our ViT-Adapter can supplement the missing high-frequency signals.

**Comparison with SETR.** Like ViTDet (Li et al., 2022b), SETR (Zheng et al., 2021) also changes the shape of features of ViT according to the task prior (see Figure 3(a)), thus allowing ViT to achieve better segmentation performance. Although this paradigm shares some similarities with our approach, *e.g.* combining ViT and convolutions, they have three main differences: (1) In addition to the task prior, our method also takes the information of the input image (the input prior) into consideration when adapting ViT to dense prediction tasks; (2) The input prior will constantly interact with ViT's features, making the output features more suitable for dense prediction tasks; (3) Our method is an adapter that is general in both detection and segmentation tasks, and moreover achieves better results than segmentation-specific head SETR (Zheng et al., 2021).

**Comparison with other Adapters.** We would like to clarify the differences between ViT-Adapter and other adapters (Jia et al., 2022; Bahng et al., 2022; Chen et al., 2022; Zhang et al., 2022; Jie & Deng, 2022) for ViTs, from two aspects as follows:

(1) *Different tasks.* Our method is designed for dense prediction tasks, while VPT (Jia et al., 2022), Visual Prompt (Bahng et al., 2022), AdaptFormer (Chen et al., 2022), NOAH (Zhang et al., 2022), and Convpass (Jie & Deng, 2022) are mainly proposed for classification tasks. By training the parameters only in input spaces, or some modules attached to the backbone, or their combination, these models perform well on classification and even obtain better results than full-tuning models.

However, when applying these methods (Jia et al., 2022; Bahng et al., 2022; Chen et al., 2022; Zhang et al., 2022; Jie & Deng, 2022) to dense prediction tasks, they perform below expectations. For example, we see from Table 11 that the performance of VPT (Jia et al., 2022) has a large gap with the baseline ViT-L (Zheng et al., 2021) on ADE20K.

| Method | Trainable Param. | mIoU (ss/ms) |
|---|---|---|
| ViT-L (full-tuning) | 318M | 48.3 / 50.1 |
| ViT-L (w/ VPT) | 16M | 44.0 / 45.6 |
| ViT-Adapter-L (full-tuning) | 332M | 52.9 / 53.7 |
| ViT-Adapter-L (frozen ViT) | 30M | 49.0 / 50.6 |

Table 11: **Comparison with VPT.**

(2) *Different targets.* These mentioned adapters (Jia et al., 2022; Bahng et al., 2022; Chen et al., 2022; Zhang et al., 2022; Jie & Deng, 2022) aim to explore parameter-efficient transfer learning, while the goal of our ViT-Adapter is to push the performance boundaries of plain ViT downstream applications, make ViT more general for downstream tasks, and efficiently utilize large-scale ViT weights pre-trained in different ways. We argue that these two technical lines are orthogonal, as shown in the last column in Table 11. Combining ViT-Adapter with these adapters to achieve efficient and accurate transfer learning of dense prediction is a research topic worth exploring.

**ViTDet's Performance.** The higher performance of the original ViTDet (Li et al., 2022b) comes from stronger training settings. Specifically, ViTDet adopts a more expensive training strategy than ours, *i.e.*, loading the MAE (He et al., 2021) pre-trained weights, and using the Large Scale Jitter (Ghiasi et al., 2021) augmentation to train the model for **100 epochs**. This setting leads to almost 3 times the training cost compared to the commonly used 36 epochs (*i.e.*, 3×+MS schedule). And to some extent, it reveals the lack of image-related inductive biases in ViT will lead to slow convergence on dense prediction tasks.

For fair comparisons, we benchmark all plain ViT detectors, including ViTDet (Li et al., 2022b) and our ViT-Adapter under the commonly used 3×+MS training schedule, and use the same ImageNet-1K pre-trained weights (*i.e.*, DeiT) as initialization. It makes sense that our ViT-Adapter achieves better performance than ViTDet under this setting, because our adapter injects image-related prior into the plain ViT, which can speed up convergence and improve performance.

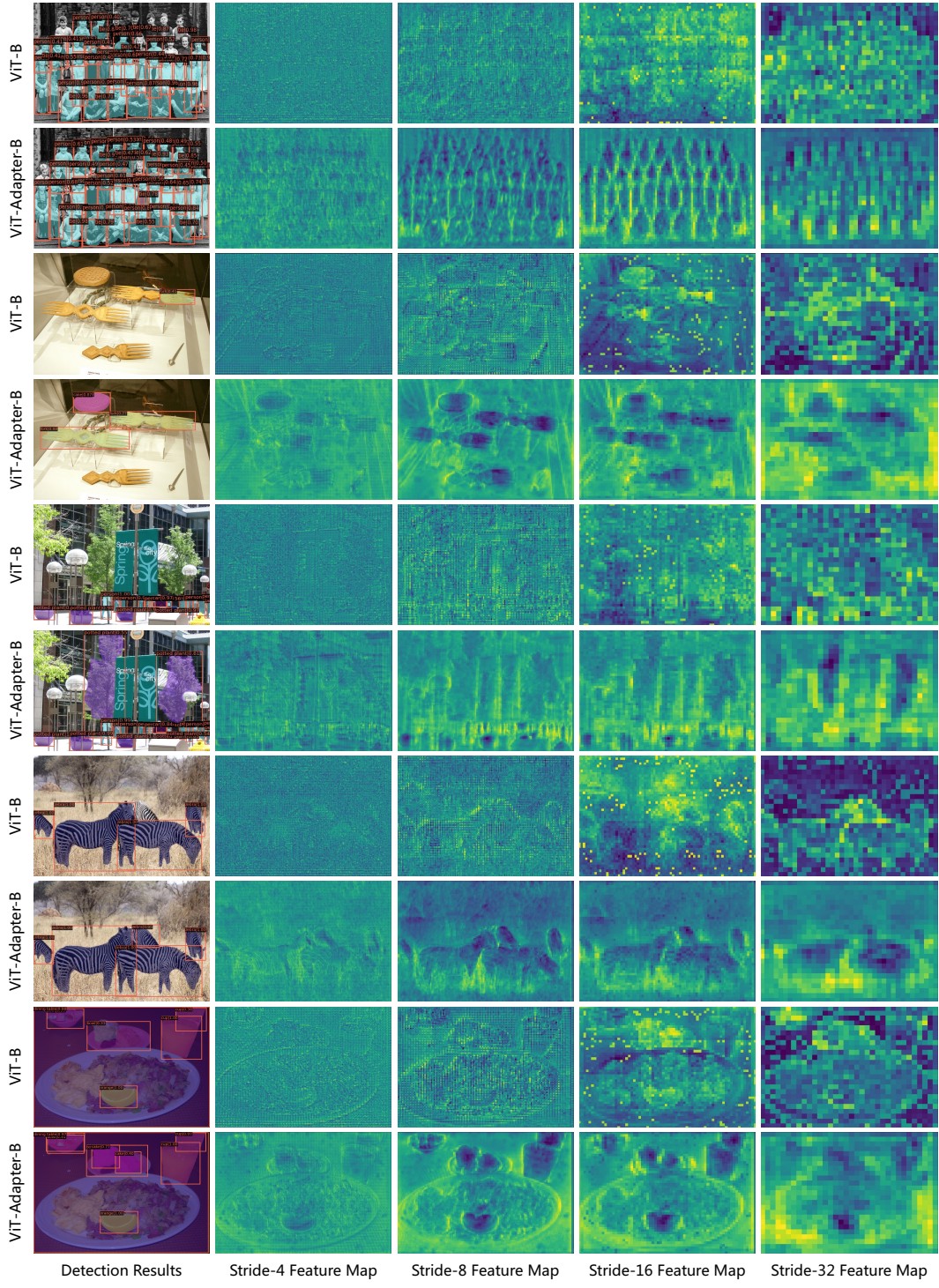

Figure 7: **Visualization of feature maps for object detection and instance segmentation.** Compared to the ViT baseline (Li et al., 2021b), our ViT-Adapter yields more fine-grained multi-scale feature maps, thus improving localization quality and reducing missed detection.

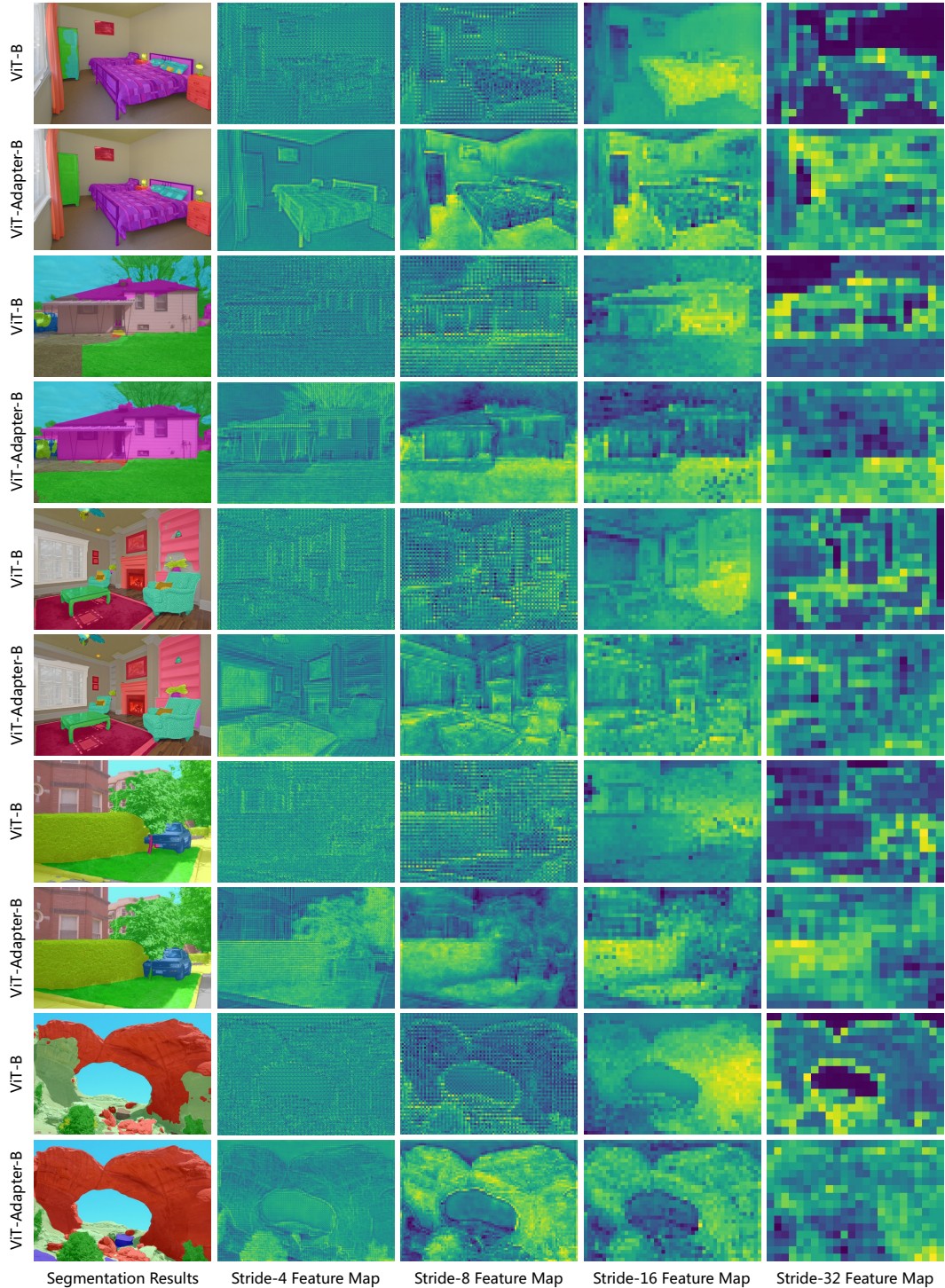

Figure 8: **Visualization of feature maps for semantic segmentation.** Compared to the ViT baseline (Li et al., 2021b), our ViT-Adapter yields more fine-grained multi-scale feature maps with rich local edges and textures, thus improving the performance of semantic segmentation.

