# OpenReview forum: "Vision Transformer Adapter for Dense Predictions"
_ICLR.cc/2023/Conference — ICLR 2023 notable top 25%_

### Official Review · Reviewer_Ycr4 · 2022-10-19

**Confidence:** 4
**Clarity, Quality, Novelty And Reproducibility:** 1. It is unclear about ViTDet's perfo…
**Correctness:** 3
**Technical Novelty And Significance:** 3
**Empirical Novelty And Significance:** 3
**Recommendation:** 6

**Strength And Weaknesses:**

#### Strength

1. Promising performance. ViT-Adapter achieves remarkable performance on several dense prediction benchmarks, using the plain ViT model.

2. The paper is clearly written and easy to follow.

3. Extensive experiments. The experiments are solid and comprehensive.



#### Weakness

1. Only the number of parameters is reported in the table. More metrics such as FLOPs and inference latency are more important.


**Summary Of The Paper:**

This paper designed a simple yet effective adapter module that helps plain ViT behave well for dense prediction tasks, including object detection, instance segmentation, and semantic segmentation. Benefited by the inductive bias introduced by the spatial prior module and feature interaction module, the plain ViT archives remarkable performance on dense prediction tasks.

**Summary Of The Review:**

See above.

---

> ### Author Response · Authors · 2022-11-19
> **Response to Reviewer Ycr4**
>
> Thanks for your constructive comments. We provide our feedback as follows.
>
> ---------------------
>
> **Q1: FLOPs and inference latency of ViT-Adapter?**
>
> **A1:** Thanks for your detailed comment. We have added a subsection in Appendix A.2 (page 18) to analyze the FLOPs and inference latency (FPS) of our method. Some results compared with representative ViT-based methods [1] on the ADE20K dataset are shown in the table below. The FPS is measured by FP16 inference using the tool in MMSegmentation with a single A100 GPU. We see that our parameters, training/inference speed, and computational/memory overhead are similar to mainstream ViT-based approaches.
>
> Although due to the speed bottleneck of the ViT backbone and deformable attention, the latency of ViT-based methods (including our ViT-Adapter) has no advantage over CNN-based methods, our method can still benefit the computer vision community with its leading accuracy, and it has already been adopted as a default framework for dense prediction in recent state-of-the-art vision foundation models [2,3].
>
> | method                      | segmentor        | pre-train | #param | #FLOPs | train time | train mem. | FPS  | mIoU (ss) | mIoU (ms) |
> | --------------------------- | ---------------- | --------- | ------ | ------ | ---------- | ---------- | ---- | --------- | --------- |
> | ViT-B                       | SETR-PUP [1]     | IN-1K     | 98M    | 170G   | 0.16s/iter | 9.5G       | 30.3 | 46.3      | 47.3      |
> | ViT-B                       | Semantic FPN [4] | IN-1K     | 98M    | 147G   | 0.15s/iter | 5.6G       | 29.7 | 46.4      | 47.6      |
> | ViT-Adapter-B               | Semantic FPN [4] | IN-1K     | 105M   | 183G   | 0.16s/iter | 7.5G       | 26.7 | **47.9**  | **48.9**  |
> | ViT-L                       | SETR-PUP [1]     | IN-22K    | 318M   | 425G   | 0.25s/iter | 16.8G      | 14.0 | 48.6      | 50.1      |
> | ViT-L                       | Semantic FPN [4] | IN-22K    | 321M   | 414G   | 0.21s/iter | 14.1G      | 15.5 | 51.5      | 52.0      |
> | ViT-Adapter-L$_\rm {light}$ | Semantic FPN [4] | IN-22K    | 324M   | 445G   | 0.23s/iter | 15.2G      | 13.5 | 52.7      | 53.5      |
> | ViT-Adapter-L               | Semantic FPN [4] | IN-22K    | 332M   | 473G   | 0.25s/iter | 16.0G      | 12.9 | **52.9**  | **53.7**  |
>
> ----------------
>
> **Q2: The original paper of ViTDet reported higher performance.**
>
> **A2:** Thanks for the detailed comment. The higher performance of the original ViTDet comes from stronger training settings.
>
> Specifically, ViTDet [5] adopts a more expensive training strategy than ours, i.e., loading the MAE [6] pre-trained weights, and using the Large Scale Jitter [7] augmentation to train the model for **100 epochs**. This setting leads to almost 3 times the training cost compared to the commonly used 36 epochs (i.e., 3x+MS schedule). And to some extent, it reveals the lack of image-related inductive bias in ViT will lead to slow convergence on dense prediction tasks.
>
> For fair comparisons, we benchmark all plain ViT detectors, including ViTDet and our ViT-Adapter under the commonly used 3x+MS training schedule, and use the same ImageNet-1K pre-trained weights (DeiT [8]) as initialization. It makes sense that our ViT-Adapter achieves better performance than ViTDet under this setting, because our adapter injects image-related prior into the plain ViT, which can speed up convergence and improve performance.
>
> We have added these explicit explanations in the appendix of our revision (page 19).
>
> ----------
>
> **Reference**
>
> [1] Zheng, Sixiao, et al. "Rethinking semantic segmentation from a sequence-to-sequence perspective with transformers." Proceedings of the IEEE/CVF conference on computer vision and pattern recognition. 2021.
>
> [2] Wang, Wenhui, et al. "Image as a foreign language: Beit pretraining for all vision and vision-language tasks." arXiv preprint arXiv:2208.10442 (2022).
>
> [3] Fang, Yuxin, et al. "EVA: Exploring the Limits of Masked Visual Representation Learning at Scale." arXiv preprint arXiv:2211.07636 (2022).
>
> [4] Kirillov, Alexander, et al. "Panoptic feature pyramid networks." Proceedings of the IEEE/CVF conference on computer vision and pattern recognition. 2019.
>
> [5] Li, Yanghao, et al. "Exploring plain vision transformer backbones for object detection." arXiv preprint arXiv:2203.16527 (2022).
>
> [6] He, Kaiming, et al. "Masked autoencoders are scalable vision learners." Proceedings of the IEEE/CVF Conference on Computer Vision and Pattern Recognition. 2022.
>
> [7] Ghiasi, Golnaz, et al. "Simple copy-paste is a strong data augmentation method for instance segmentation." Proceedings of the IEEE/CVF Conference on Computer Vision and Pattern Recognition. 2021.
>
> [8] Touvron, Hugo, et al. "Training data-efficient image transformers & distillation through attention." International Conference on Machine Learning. PMLR, 2021.

---

> > ### Comment · Reviewer_Ycr4 · 2022-11-19
> > **Further concerns.**
> >
> > Thanks for your reply.
> >
> > For question 2, I was curious about the performance of ViT-Adapter under ViTDet's default setting.
> >
> > Whether does ViT-Adapter still outperform ViTDet or not?

---

> > > ### Author Response · Authors · 2022-11-19
> > > **Response to Reviewer Ycr4's further concerns**
> > >
> > > Thanks for your quick reply. As shown in the table below, although it's not a perfectly controlled comparison, our method obtains similar performance on COCO val2017 while taking fewer training epochs (36 vs. 100) and GPU hours (about 40%) than ViTDet. We argue that under ViTDet's default setting, ViT-Adapter will still outperform ViTDet. We will conduct experiments to verify this, which will take a few days.
> > >
> > > | method        | detector               | pre-train   | epoch | box AP | mask AP |
> > > | ------------- | ---------------------- | ---------- | ----- | ------ | ------- |
> > > | ViTDet-L [1]  | Cascade Mask R-CNN [2] | MAE [3]    | 100   | 60.4   | 52.2    |
> > > | ViT-Adapter-L  | Cascade Mask R-CNN [2] | MAE [3]    | 100   | 60.9   | 52.6    |
> > > | ViT-Adapter-L  | HTC++ [4]              | BEiTv2 [5] | 36    | 60.5  | 52.5    |
> > >
> > > ----------------
> > >
> > > **Reference**
> > >
> > > [1] Li, Yanghao, et al. "Exploring plain vision transformer backbones for object detection." arXiv preprint arXiv:2203.16527 (2022).
> > >
> > > [2] Cai, Zhaowei, and Nuno Vasconcelos. "Cascade R-CNN: high quality object detection and instance segmentation." IEEE transactions on pattern analysis and machine intelligence 43.5 (2019): 1483-1498.
> > >
> > > [3] He, Kaiming, et al. "Masked autoencoders are scalable vision learners." Proceedings of the IEEE/CVF Conference on Computer Vision and Pattern Recognition. 2022.
> > >
> > > [4]
> > > Liu, Ze, et al. "Swin transformer: Hierarchical vision transformer using shifted windows." Proceedings of the IEEE/CVF International Conference on Computer Vision. 2021.
> > >
> > > [5] Peng, Zhiliang, et al. "Beit v2: Masked image modeling with vector-quantized visual tokenizers." arXiv preprint arXiv:2208.06366 (2022).

---

> > > > ### Author Response · Authors · 2022-11-25
> > > > **Response to Reviewer Ycr4's further concerns (Part 2)**
> > > >
> > > > Thanks for waiting. Our experiment using ViTDet's default setting has been completed. As
> > > > shown in the table above (row 2), under the default setting of ViTDet, our ViT-Adapter-L achieves
> > > > 60.9 box AP and 52.6 mask AP on COCO val2017, which are +0.5 box AP (60.9 vs 60.4) and +0.4
> > > > mask AP (52.6 vs 52.2) higher than ViTDet-L, respectively.
> > > >
> > > > These results show that our method can not only achieve competitive performance with short training schedules (e.g. 36 epochs), but also maintain good performance for a long training schedule of 100 epochs.

---

### Official Review · Reviewer_hMeh · 2022-10-22

**Confidence:** 4
**Clarity, Quality, Novelty And Reproducibility:** The clarity, quality, novelty and rep…
**Correctness:** 3
**Technical Novelty And Significance:** 3
**Empirical Novelty And Significance:** 3
**Recommendation:** 5

**Strength And Weaknesses:**

Strengths:
+ The paper is written-well and easy to follow, the idea is intuitive and clear.
+ The experiments are solid and entensive.
+ The authors have made good efforts to ablate different components of their approach.
+ The performances improvement of the proposed method are impressed.

Weaknesses:
- Adapters in ViT is an interesting research topic, there were some works focused on it  [a-e]. I strongly encourage authors to take them into deep discussion. For example, the differences and advantages between them. The additional fair experiments should be  also used for discussion.

[a] "Visual prompt tuning." In ECCV, 2022.

[b] B"Exploring visual prompts for adapting large-scale models." arXiv, 2022.

[c] "AdaptFormer: Adapting Vision Transformers for Scalable Visual Recognition." In NeurIPS, 2022.

[d]  "Neural Prompt Search." arXiv, 2022.

[e] "Convolutional bypasses are better vision transformer adapters." arXiv, 2022.

- Authors inject spatial priors and multi-scale feautres learning into ViT and achieves significant performances. I doubt that can we add the other priors or learning tricks into ViT to achieve better performances. That is, the module in Vit-Adapter can be replaced by others that is important for dense prediction tasks? Can authors provide additional ablation studies on it.

**Summary Of The Paper:**

This paper proposed ViT-Adapter for dense prediction task. Essentially, Vit-Adapter utilizes additional modules to inject the image prior into ViT, encouraging ViT to learn transferable representations for the specific dense prediction tasks. The extensive experiments on several dense prediction tasks (e.g., object detection, instanve segmentation, and semantic segmentation) demonstrate the effectiveness of the proposed method.

**Summary Of The Review:**

This work is interesting and written-well, but some of concerns needed to be improved. I'm willing to raise my rating score if authors carefully tackle my concerns.

---

> ### Author Response · Authors · 2022-11-19
> **Response to Reviewer hMeh**
>
> Thanks for your constructive comments. We provide our feedback as follows.
>
> -------------------
>
> **Q1: The differences and advantages between ViT-Adapter and other adapters for ViT.**
>
> **A1:** Thanks for your comment. We would like to clarify the differences between ViT-Adapter and other adapters [1-5] for ViTs, from two aspects as follows:
>
> (1) *Different tasks.* Our method is designed for dense prediction tasks, while VPT [1],  Visual Prompt [2], AdaptFormer [3], NOAH [4], and Convpass [5] are mainly proposed for classification tasks. By training the parameters only in input spaces [1,2], or some modules attached to the backbone [3,5], or their combination [4], these models have good performance on classification tasks and even obtain better results than full tuning models.
>
> However, when applying these methods [1-5] to dense prediction tasks, they perform below expectations. For example, we see from the table below that the performance of VPT [1] has a large gap with the baseline ViT-L [6] on ADE20K.
>
> | Method                      | Trainable Param | ADE20K mIoU (ss/ms) |
> | --------------------------- | --------------- | ------------------- |
> | ViT-L (full tuning) [6]     | 318M            | 48.3/50.1           |
> | ViT-L (VPT) [1, 6]          | 16M             | 44.0/45.6           |
> | ViT-Adapter-L (full tuning) | 332M            | 52.9/53.7           |
> | ViT-Adapter-L (freeze ViT)  | 30M             | 49.0/50.6           |
>
>
> (2) *Different targets.* These mentioned adapters [1-5] aim to explore parameter-efficient transfer learning, while the goal of our ViT-Adapter is to push the performance boundaries of plain ViT downstream applications, make ViT more general for downstream tasks, and efficiently utilize large-scale ViT weights pre-trained in different ways. We argue that these two technical lines are orthogonal, as shown in the last column of the table above. Combining ViT-Adapter with these adapters [1-5] to achieve efficient and accurate transfer learning of dense prediction is a research topic worth exploring.
>
> These discussions have been added in Appendix A.2 (page 18).
>
> ------------------------
>
> **Q2: Can we add the other priors or learning tricks into ViT to achieve better performances?**
>
> **A2:** As the prior information inner model plays an important role in dense prediction tasks, our work provides an effective way of injecting the task prior and input prior into plain ViT. For the common detection and segmentation tasks, we argue that spatial prior and hierarchical features are the most commonly used prior information lacking in plain ViT. With extensive experiments to introduce these two prior, we demonstrate the effectiveness of our method, and the visualizations in Fig. 5 also properly support the concept of our method.
> But for some more specific downstream tasks, such as edge detection or video depth estimation, introducing the more carefully designed prior like edge or optical flow prior may have a positive effect. We'd like to leave it for future exploration.
>
> ----------------
>
> **Reference**
>
>
> [1] "Visual prompt tuning." In ECCV, 2022.
>
> [2] "Exploring visual prompts for adapting large-scale models." arXiv, 2022.
>
> [3] "AdaptFormer: Adapting Vision Transformers for Scalable Visual Recognition." In NeurIPS, 2022.
>
> [4] "Neural Prompt Search." arXiv, 2022.
>
> [5] "Convolutional bypasses are better vision transformer adapters." arXiv, 2022.
>
> [6] "Rethinking semantic segmentation from a sequence-to-sequence perspective with transformers." In CVPR, 2021.

---

### Official Review · Reviewer_SVHK · 2022-10-25

**Confidence:** 3
**Clarity, Quality, Novelty And Reproducibility:** The clarity is clear. The novelty and…
**Correctness:** 3
**Technical Novelty And Significance:** 3
**Empirical Novelty And Significance:** 3
**Recommendation:** 8

**Strength And Weaknesses:**

Strength:
+ Applying the plain ViT to downstream tasks is an improtant and meanful topic. The motivation is well explained and makes sense. The proposed technique is simple yet effectiveness.
+ The paper is well-written and well-organized. The figures and visualizations are clear and easy to follow.
+ The experiments and ablation studies are comprehensive. The proposed adapter shows significant superiority over the previous works.
And it achieves state-of-the-art performance on several tasks.

Weakness:
- It would be better to give the computational complexity and latency in the experiments.

**Summary Of The Paper:**

The application of plain ViT on dense prediction tasks suffers from unacceptable complexity. The paper proposes an injection adapter with multi-scale features to use a well-pretrained ViT model and transfer it to downstream task. The experiments are conducted on object detection, instance segmentation, and semantic segmentation. The proposed technique achieves state-of-the-art on COCO dataset, which well demonstrate the high effectiveness.

**Summary Of The Review:**

This paper explores an important problem in multimodality and proposes a simple yet effective solution to transfer the ViT models to dense prediction tasks. The experiments give strong evidences to well support.

---

> ### Author Response · Authors · 2022-11-19
> **Response to Reviewer SVHK**
>
> Thanks for your constructive comments. We provide our feedback as follows.
>
> ------------------------
> **Q1: FLOPs and latency of ViT-Adapter?**
>
> **A1:**  Thanks for your comment. We have added a subsection in Appendix A.2 (page 18) to analyze the FLOPs and latency of our method. Some results compared with the representative ViT-based method SETR [1] on the ADE20K dataset are shown in the table below. The FPS is measured by FP16 inference using the tool in MMSegmentation with a single A100 GPU. We see that the parameters, training/inference speed, and computational/memory overhead are similar to mainstream ViT-based approaches.
>
> Although due to the speed bottleneck of the ViT backbone and deformable attention, the latency of ViT-based methods (including our ViT-Adapter) has no advantage over CNN-based methods, our method can still benefit the computer vision community with its leading accuracy, and it has already been adopted as a default framework for dense prediction in recent state-of-the-art vision foundation models [2,3].
>
> | method                      | segmentor        | pre-train | #param | #FLOPs | train time | train mem. | FPS  | mIoU (ss) | mIoU (ms) |
> | --------------------------- | ---------------- | --------- | ------ | ------ | ---------- | ---------- | ---- | --------- | --------- |
> | ViT-B                       | SETR-PUP [1]     | IN-1K     | 98M    | 170G   | 0.16s/iter | 9.5G       | 30.3 | 46.3      | 47.3      |
> | ViT-B                       | Semantic FPN [4] | IN-1K     | 98M    | 147G   | 0.15s/iter | 5.6G       | 29.7 | 46.4      | 47.6      |
> | ViT-Adapter-B               | Semantic FPN [4] | IN-1K     | 105M   | 183G   | 0.16s/iter | 7.5G       | 26.7 | **47.9**  | **48.9**  |
> | ViT-L                       | SETR-PUP [1]     | IN-22K    | 318M   | 425G   | 0.25s/iter | 16.8G      | 14.0 | 48.6      | 50.1      |
> | ViT-L                       | Semantic FPN [4] | IN-22K    | 321M   | 414G   | 0.21s/iter | 14.1G      | 15.5 | 51.5      | 52.0      |
> | ViT-Adapter-L$_\rm {light}$ | Semantic FPN [4] | IN-22K    | 324M   | 445G   | 0.23s/iter | 15.2G      | 13.5 | 52.7      | 53.5      |
> | ViT-Adapter-L               | Semantic FPN [4] | IN-22K    | 332M   | 473G   | 0.25s/iter | 16.0G      | 12.9 | **52.9**  | **53.7**  |
>
> -------------------
>
> **Reference**
>
> [1] Zheng, Sixiao, et al. "Rethinking semantic segmentation from a sequence-to-sequence perspective with transformers." Proceedings of the IEEE/CVF conference on computer vision and pattern recognition. 2021.
>
> [2] Wang, Wenhui, et al. "Image as a foreign language: Beit pretraining for all vision and vision-language tasks." arXiv preprint arXiv:2208.10442 (2022).
>
> [3] Fang, Yuxin, et al. "EVA: Exploring the Limits of Masked Visual Representation Learning at Scale." arXiv preprint arXiv:2211.07636 (2022).
>
> [4] Kirillov, Alexander, et al. "Panoptic feature pyramid networks." Proceedings of the IEEE/CVF conference on computer vision and pattern recognition. 2019.

---

### Official Review · Reviewer_BCxb · 2022-11-02

**Confidence:** 5
**Correctness:** 4
**Technical Novelty And Significance:** 3
**Empirical Novelty And Significance:** 3
**Recommendation:** 8

**Clarity, Quality, Novelty And Reproducibility:**

The idea itself is not very novel, but the proposed architecture is interesting and have been optimized  for different applications. The code is supposed to be published on GitHub based on author's claims, hence it should be reproducible.

**Strength And Weaknesses:**

########## Strength

1. The paper is well-written and easy to follow. Visualizations properly demonstrate the concept of the proposed architecture.

2. The overall design of the ViT-adapter architecture is quite general and can be used in different applications. Especially, using conv-based prior modules seem to be effective and play an important role. Unlike the previous ViTDet model, the adapter seems to also make use of the input image which can be effectively leveraged along with intermediate extracted features. In addition, the idea of using both conv and transformer-based (i.e. encoders with MLP and cross attention) layers in the adapter seems to be more optimal than using conv-based pyramid-like architectures.

3. The method has been thoroughly tested on many different tasks (both supervised and self-supervised) and achieve SOTA performance on COCO test-dev dateset --although there seems to be other approaches such as FD-SwinV2-G [1], DINO [2], etc. to have better performance, but the performance is still quite competitive.

########## Weakness

1. Despite the effective design of the proposed architecture, the idea of the ViT-adapter is not very new. The first instance of such works was SETR [3] which simply used a CNN-based decoder for feature processing in segmentation.

2. As the name indicates, the architecture is solely designed for ViT architectures, and this may limit its applicability as hierarchical models such as Swin Transformer have gained a lot interest. It is fair to say that with minimal changes, the adapter could be used with Swin and other similar architectures. Hence it could be beneficial to study whether it could be useful when the feature extractor is more powerful.

3. How does the adapter affect the throughput and memory foot-print ? I am concerned that it may not be feasible to use it in practical applications due to it complex architecture. It would be nice to have comparison to SETR and ViTDet for throughput.


[1]: Wei, Y., Hu, H., Xie, Z., Zhang, Z., Cao, Y., Bao, J., Chen, D. and Guo, B., 2022. Contrastive Learning Rivals Masked Image Modeling in Fine-tuning via Feature Distillation. arXiv preprint arXiv:2205.14141.

[2]: Zhang, H., Li, F., Liu, S., Zhang, L., Su, H., Zhu, J., Ni, L.M. and Shum, H.Y., 2022. Dino: Detr with improved denoising anchor boxes for end-to-end object detection. arXiv preprint arXiv:2203.03605.

[3]: Zheng, S., Lu, J., Zhao, H., Zhu, X., Luo, Z., Wang, Y., Fu, Y., Feng, J., Xiang, T., Torr, P.H. and Zhang, L., 2021. Rethinking semantic segmentation from a sequence-to-sequence perspective with transformers. In Proceedings of the IEEE/CVF conference on computer vision and pattern recognition (pp. 6881-6890).

**Summary Of The Paper:**

This work attempts to address the problem of lack of inductive bias in plain ViT architectures which hinders its performance for dense prediction tasks such as semantic segmentation and object detection. Specifically, the authors propose an adapter architecture that can extract and learn representations that are useful for dense prediction tasks. The proposed methodology has been trained and tested on different down-stream tasks and purportedly achieves competitive performance on COCO test-dev dateset in terms of box AP and mask AP.

**Summary Of The Review:**

The proposed model addresses the problem of lack of inductive bias in ViT architecture and has been shown to be effective in many different application. The model has been testes rigorously on different dataset and achieves competitive performance on COCO test-dev dateset. As a result, I believe these contributions merit acceptance.

---

> ### Author Response · Authors · 2022-11-19
> **Response to Reviewer BCxb (Part1)**
>
> Thanks for your positive comments! We hope ViT-Adapter can provide good practices for applying plain ViT in dense prediction tasks, and motivate researchers to make effective use of pre-trained ViTs in different ways. All the code and models will be available.
>
> ---------
> **Q1: The differences and relationships between SETR and ViT-Adapter.**
>
> **A1:**  Like ViTDet [1], SETR [2] also changes the shape of features of ViT according to the task prior (see Fig. 3 (a)), thus allowing ViT to achieve better segmentation performance. Although this paradigm shares some similarities with our approach, e.g. combining ViT and convolutions, they have three main differences:
>
> (1) In addition to the task prior, our method also takes the information of the input image (the input prior) into consideration when adapting ViT to dense prediction tasks;
>
> (2) The input prior will constantly interact with ViT's features, making the output features more suitable for dense prediction tasks;
>
> (3) Our method is an adapter that is general in both detection and segmentation tasks, and moreover achieves better results than segmentation-specific head SETR [2].
>
> We have added this discussion in Appendix A.2.
>
> -------------------
>
> **Q2: Could the adapter be used with Swin and other similar architectures?**
>
> **A2:** Yes, but the performance gain is not as significant as ViT. With some small modifications, the adapter could work with hierarchical vision transformers such as Swin [3]. We apply our adapter to Swin-T for object detection, and train the model with Mask R-CNN [4] with the 3x training schedule. As shown in the table below, our adapter improves the performance of Swin-T by +0.7 box AP and +0.5 mask AP, and this improvement is less than that (+4.2 box AP and  +2.9 mask AP) of ViT. This result is within our expectations, since Swin has *already* included many inductive biases needed for dense prediction tasks, such as shifted local windows and the hierarchical structure.
>
>
> | method               | #param | #FLOPs | train time | box AP      | mask AP     |
> | -------------------- | ------ | ------ | ---------- | ----------- | ----------- |
> | Swin-T [3]           | 48M    | 267G   | 0.27s/iter | 46.0        | 41.6        |
> | Swin-T + our adapter | 50M    | 293G   | 0.33s/iter | 46.7 (+0.7) | 42.1 (+0.5) |
>
> ----------------------
> **Q3: How does the adapter affect the throughput and memory footprint? I am concerned that it may not be feasible to use it in practical applications.**
>
> **A3:** Thanks for your detailed review. We have added a subsection in Appendix A.2 (page 18) to compare the computational efficiency with representative ViT-based methods SETR [2] (see Table 11) and ViTDet [1] (see Table 12). Some results compared with SETR on the ADE20K dataset are shown in the table below. The FPS is measured by FP16 inference using the tool in MMSegmentation with a single A100 GPU. We see that our parameters, training/inference speed, and computational/memory overhead are similar to mainstream ViT-based approaches.
>
> | method                      | segmentor        | pre-train | #param | #FLOPs | train time | train mem. | FPS  | mIoU (ss) | mIoU (ms) |
> | --------------------------- | ---------------- | --------- | ------ | ----- | ---------- | ---------- | ---- | --------- | --------- |
> | ViT-B                       | SETR-PUP [2]     | IN-1K     | 98M    | 170G  | 0.16s/iter | 9.5G       | 30.3 | 46.3      | 47.3      |
> | ViT-B                       | Semantic FPN [5] | IN-1K     | 98M    | 147G  | 0.15s/iter | 5.6G       | 29.7 | 46.4      | 47.6      |
> | ViT-Adapter-B               | Semantic FPN [5] | IN-1K     | 105M   | 183G  | 0.16s/iter | 7.5G       | 26.7 | **47.9**  | **48.9**  |
> | ViT-L                       | SETR-PUP [2]     | IN-22K    | 318M   | 425G  | 0.25s/iter | 16.8G      | 14.0 | 48.6      | 50.1      |
> | ViT-L                       | Semantic FPN [5] | IN-22K    | 321M   | 414G  | 0.21s/iter | 14.1G      | 15.5 | 51.5      | 52.0      |
> | ViT-Adapter-L$_{\rm light}$ | Semantic FPN [5] | IN-22K    | 324M   | 445G  | 0.23s/iter | 15.2G      | 13.5 | 52.7      | 53.5      |
> | ViT-Adapter-L               | Semantic FPN [5] | IN-22K    | 332M   | 473G  | 0.25s/iter | 16.0G      | 12.9 | **52.9**  | **53.7**  |

---

> > ### Author Response · Authors · 2022-11-19
> > **Response to Reviewer BCxb (Part2)**
> >
> > In addition, our method has the following potential applications:
> >
> > (1) The proposed method has leading performance on both detection and segmentation tasks, and even successfully pushes the performance boundaries of popular benchmarks such as ADE20K, COCO-Stuff, and Pascal Context. In addition, recently proposed state-of-the-art ViT-based foundation models, such as BEiT-3 [6] and EVA [7], adopt our method as the default segmentation framework.
> >
> > (2) Due to its high performance, it can also serve as a teacher model to distill real-time small models for practical deployment.
> >
> > (3) As the model for downstream tasks with real-time requirements, the speed bottleneck of our method lies in the ViT backbone and deformable attention, which is also the main problem in the deployment of the transformer-based model at present. We will continue to explore this problem in our follow-up versions.
> >
> > ----------------------
> >
> > **Reference**
> >
> > [1] Li, Yanghao, et al. "Exploring plain vision transformer backbones for object detection." arXiv preprint arXiv:2203.16527 (2022).
> >
> > [2] Zheng, Sixiao, et al. "Rethinking semantic segmentation from a sequence-to-sequence perspective with transformers." Proceedings of the IEEE/CVF conference on computer vision and pattern recognition. 2021.
> >
> > [3] Liu, Ze, et al. "Swin transformer: Hierarchical vision transformer using shifted windows." Proceedings of the IEEE/CVF International Conference on Computer Vision. 2021.
> >
> > [4] He, Kaiming, et al. "Mask r-cnn." Proceedings of the IEEE international conference on computer vision. 2017.
> >
> > [5] Kirillov, Alexander, et al. "Panoptic feature pyramid networks." Proceedings of the IEEE/CVF conference on computer vision and pattern recognition. 2019.
> >
> > [6] Wang, Wenhui, et al. "Image as a foreign language: Beit pretraining for all vision and vision-language tasks." arXiv preprint arXiv:2208.10442 (2022).
> >
> > [7] Fang, Yuxin, et al. "EVA: Exploring the Limits of Masked Visual Representation Learning at Scale." arXiv preprint arXiv:2211.07636 (2022).

---

> > > ### Comment · Reviewer_BCxb · 2022-11-19
> > > **Response to Authors**
> > >
> > > I appreciate the author's efforts in addressing my concerns.
> > >
> > > In addition, I would like to request if the authors may commit to releasing the code repository for their work upon it acceptance, including all the new pieces (e.g. Swin Transformers, etc.) . I believe this would be helpful for the community.

---

> > > > ### Author Response · Authors · 2022-11-20
> > > > **Response to Reviewer BCxb**
> > > >
> > > > Thank you for your appreciation of our work! we are willing to release the code and model of our method, including these new pieces like Swin Transformer + adapter.

---

### Decision · Program_Chairs · 2023-01-20

**Decision:**

Accept: notable-top-25%

**Justification For Why Not Higher Score:**

N/A

**Justification For Why Not Lower Score:**

N/A

**Metareview: Summary, Strengths And Weaknesses:**

This submission receives 3 positive reviews and 1 slightly negative review.  The raised concerns are the difference between prior adapter works. In the rebuttal phase, the authors have shown the current design is for dense prediction with favorable performance. After checking all the reviews, the AC feels this is a well fit for this venue and recommends acceptance.

**Note From Pc:**

if the above contains the word "oral" or "spotlight" please see: "oral" presentation means -> notable-top-5% and "spotlight" means -> notable-top-25%. As stated in our emails, we are disassociating presentation type from AC recommendations